# Jacobian-Aware Posterior Sampling for Inverse Problems

**Liav Hen**[*]                                                            *liavhen@gmail.com*
*Tel Aviv University, Israel*

**Tom Tirer**                                                             *tirer.tom@gmail.com*
*Bar-Ilan University, Israel*

**Raja Giryes**                                                          *raja@tauex.tau.ac.il*
*Tel Aviv University, Israel*

**Shady Abu-Hussein**                                                 *shady.abh@gmail.com*
*University of Cambridge, UK*

**Reviewed on OpenReview:** *https://openreview.net/forum?id=m63GJnhIN2*

## Abstract

Diffusion models provide powerful generative priors for solving inverse problems by sampling from a posterior distribution conditioned on corrupted measurements. Existing methods primarily follow two paradigms: direct methods, which approximate the likelihood term, and proximal methods, which incorporate intermediate solutions satisfying measurement constraints into the sampling process. Under standard Gaussian approximations and locally-linear measurements, we demonstrate that these approaches differ fundamentally in their treatment of the diffusion denoiser's Jacobian within the likelihood term. While this Jacobian encodes critical prior knowledge of the data distribution, training-induced non-idealities can degrade performance in zero-shot settings. In this work, we bridge direct and proximal approaches by proposing a principled **J**acobian-**A**ware **P**osterior **S**ampler (JAPS). JAPS leverages the Jacobian's prior knowledge while mitigating its detrimental effects through a corresponding proximal solution, requiring no additional computational cost. Additionally, we integrate our guidance into DDIM sampling, with a corrected conditional factor that has been missing in previous works. Our method enhances reconstruction quality across diverse linear and nonlinear noisy imaging tasks, outperforming existing diffusion-based baselines in perceptual quality while maintaining or improving distortion metrics.

## 1 Introduction

Image restoration aims to recover a high-quality image $\mathbf{x} \in \mathbb{R}^n$ from a degraded observation $\mathbf{y} = \mathcal{A}(\mathbf{x}) + \boldsymbol{\varepsilon}$, where $\mathcal{A}$ is a measurement operator and $\boldsymbol{\varepsilon}$ is additive noise. Common restoration tasks include denoising, deblurring, super-resolution and inpainting, which can also benefit high-level downstream tasks (Turgeman and Tirer, 2026). Since these inverse problems are typically ill-posed, incorporating strong structural priors is essential for accurate recovery.

While task-specific deep neural networks (DNNs) offer high performance, they often suffer from performance degradation when test-time observations deviate from training assumptions (Hussein et al., 2020; Shocher et al., 2018; Tirer and Giryes, 2019). To address this, zero-shot frameworks such as Plug-and-Play (PnP) and Regularization by Denoising (RED) (Romano et al., 2017; Tirer and Giryes, 2018; Venkatakrishnan et al., 2013; Zhang et al., 2017) leverage pretrained denoisers as priors. More recently, diffusion models (Ho et al., 2020; Song and Ermon, 2019; Song et al., 2020b) have emerged as powerful generative priors, enabling

---

[*]Corresponding author: liavhen@gmail.com
  Code available at `https://github.com/liavhen/JAPS`

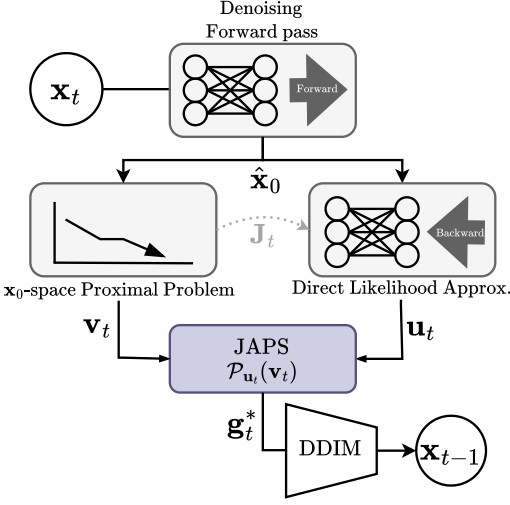

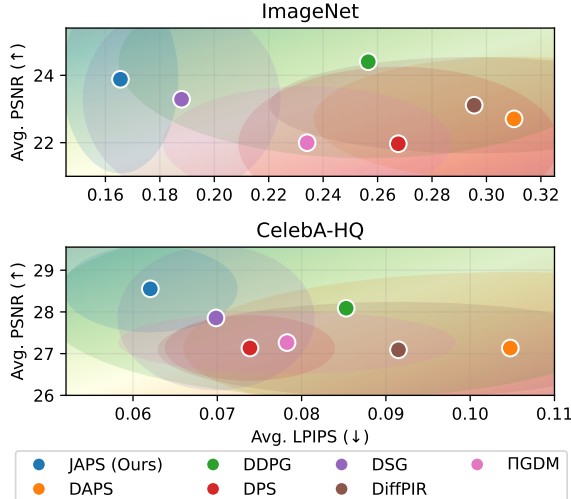

(a) **Schematic overview of our approach.** We bridge between direct and proximal methods, which, under a linear-Gaussian framework, essentially differ by the presence of the denoiser's Jacobian, leveraging both to formulate an enhanced likelihood surrogate.

(b) **Summary of experimental results.** Avg. PSNR/LPIPS across various reconstruction tasks— SR×4, Gaussian and motion deblurring, and box/random inpainting—with shaded ellipsoids for standard deviation.

Figure 1: **Proposed approach and experimental summary. (a)** We introduce an enhanced likelihood surrogate by bridging direct and proximal methods via the denoiser's Jacobian. **(b)** Performance comparison across various inverse problems, demonstrating the efficacy of our approach.

reconstructions that are both perceptually high-fidelity and consistent with measurements (Abu-Hussein et al., 2022; Chung et al., 2022; 2023; Kawar et al., 2022; Mardani et al., 2023; Song et al., 2023; 2021; Wang et al., 2023; Zhu et al., 2023; Garber and Tirer, 2024; 2025).

Existing diffusion-based posterior samplers primarily follow two paradigms: **direct methods**, which approximate the log-likelihood gradient $\nabla_{\mathbf{x}} \log p(\mathbf{y}|\mathbf{x})$ along the sampling process, and **proximal methods**, which incorporate intermediate solutions satisfying measurement constraints (Kawar et al., 2022; Chung et al., 2023; Zhu et al., 2023; Garber and Tirer, 2024; Yang et al., 2024). We show that within a framework utilizing a common isotropic Gaussian approximation and linear (or locally linear) measurements, these paradigms differ fundamentally in their treatment of the diffusion denoiser's Jacobian within the likelihood term. While this Jacobian encodes critical local geometry of the data distribution, training-induced non-idealities often lead to artifacts when used directly in zero-shot settings.

In this work, we propose **J**acobian-**A**ware **P**osterior **S**ampling (JAPS). JAPS utilizes the Jacobian's prior information while mitigating detrimental effects through a corresponding proximal estimate, without additional computational cost. Furthermore, we extend this approach to nonlinear problems and provide a mathematical framework for incorporating likelihood terms into DDIM sampling (Song et al., 2020a).

Our main contributions are summarized as follows:

**(1) Jacobian-Aware Posterior Sampling:** We identify the role of the Jacobian in posterior sampling and introduce JAPS, a sampler that bridges direct and proximal solutions to improve reconstruction quality without extra computational overhead.

**(2) Nonlinear extension:** We extend our framework to nonlinear inverse problems, demonstrating improved performance across representative nonlinear tasks compared to existing baselines.

**(3) DDIM reformulation for conditional guidance:** We provide a formulation to incorporate the log-likelihood gradient into the DDIM sampler via a conditional noise estimator. This approach preserves DDIM's original scheduling and scales with time re-spacing and varying levels of stochasticity.

## 2 Background

### 2.1 Inverse Problems

We consider the general inverse problem of recovering a high-quality image $\mathbf{x} \in \mathbb{R}^n$ from a degraded observation $\mathbf{y} = \mathcal{A}(\mathbf{x}) + \boldsymbol{\varepsilon}$, where $\mathcal{A} : \mathbb{R}^n \to \mathbb{R}^m$ is a measurement operator and $\boldsymbol{\varepsilon} \sim \mathcal{N}(\mathbf{0}, \sigma_y^2 \mathbf{I}_m)$ is additive white Gaussian noise. From a Bayesian perspective, $\mathbf{x}$ is a random vector drawn from a prior $p(\mathbf{x})$. While maximizing the likelihood $p(\mathbf{y}|\mathbf{x}) \propto \exp(-\frac{1}{2\sigma_y^2}\|\mathcal{A}(\mathbf{x}) - \mathbf{y}\|_2^2)$ is often insufficient, maximizing the posterior $p(\mathbf{x}|\mathbf{y}) \propto p(\mathbf{y}|\mathbf{x})p(\mathbf{x})$ incorporates structural priors that improve the reconstruction. A central challenge lies in utilizing an accurate prior $p(\mathbf{x})$ that captures the true data distribution.

### 2.2 Diffusion Models

Diffusion models synthesize data by reversing a gradual noising process (Ho et al., 2020; Song et al., 2020b). A forward SDE, defined by some proper functions $\mathbf{f}(\mathbf{x}, t)$ and $g(t)$, progressively corrupts clean data until, at terminal time $T$, the distribution is approximately Gaussian; generation then amounts to solving the corresponding reverse-time SDE, which requires the *score function* $\nabla_{\mathbf{x}} \log p_t(\mathbf{x})$ (Anderson, 1982):

$$d\mathbf{x} = \Big[\mathbf{f}(\mathbf{x}, t) - g(t)^2 \nabla_{\mathbf{x}} \log p_t(\mathbf{x})\Big]dt + g(t)\, d\bar{\mathbf{w}}. \tag{1}$$

In the variance-preserving (VP) formulation, the noisy state is $\mathbf{x}_t = \sqrt{\bar{\alpha}_t}\mathbf{x}_0 + \sqrt{1 - \bar{\alpha}_t}\boldsymbol{\epsilon}$, with $\boldsymbol{\epsilon} \sim \mathcal{N}(\mathbf{0}, \mathbf{I})$ and $\bar{\alpha}_t$ being the noise schedule. A neural network $\boldsymbol{\epsilon}_\theta(\mathbf{x}_t, t)$ is trained to predict $\boldsymbol{\epsilon}$, providing a score estimate:

$$\nabla_{\mathbf{x}_t} \log p_t(\mathbf{x}_t) \approx -\frac{1}{\sqrt{1 - \bar{\alpha}_t}}\boldsymbol{\epsilon}_\theta(\mathbf{x}_t, t). \tag{2}$$

Via Tweedie's formula (Efron, 2011), this network yields an estimate of the clean image:

$$\hat{\mathbf{x}}_0(\mathbf{x}_t, t) = \frac{\mathbf{x}_t - \sqrt{1 - \bar{\alpha}_t}\boldsymbol{\epsilon}_\theta(\mathbf{x}_t, t)}{\sqrt{\bar{\alpha}_t}} \approx \mathbb{E}[\mathbf{x}_0 \mid \mathbf{x}_t]. \tag{3}$$

For practical sampling, we employ the DDIM algorithm (Song et al., 2020a) for any $t \in [T, T - 1, ...1]$:

$$\mathbf{x}_{t-1} = \sqrt{\bar{\alpha}_{t-1}}\hat{\mathbf{x}}_0(\mathbf{x}_t, t) + \sqrt{1 - \bar{\alpha}_{t-1} - \sigma_t^2}\boldsymbol{\epsilon}_\theta(\mathbf{x}_t, t) + \sigma_t\boldsymbol{\epsilon}_t, \tag{4}$$

where $\boldsymbol{\epsilon}_t \sim \mathcal{N}(\mathbf{0}, \mathbf{I})$ and $\sigma_t = \eta\sqrt{1 - \frac{\bar{\alpha}_t}{\bar{\alpha}_{t-1}}}\sqrt{\frac{1 - \bar{\alpha}_{t-1}}{1 - \bar{\alpha}_t}}$, with $\eta \in [0, 1]$, determines the level of stochasticity.

### 2.3 Posterior Sampling

Zero-shot diffusion-based solvers for inverse problems approximate the posterior score $\nabla_{\mathbf{x}_t} \log p_t(\mathbf{x}_t|\mathbf{y})$, which decomposes via Bayes' rule as

$$\nabla_{\mathbf{x}_t} \log p_t(\mathbf{x}_t|\mathbf{y}) = \underbrace{\nabla_{\mathbf{x}_t} \log p_t(\mathbf{x}_t)}_{\text{prior score}} + \underbrace{\nabla_{\mathbf{x}_t} \log p_t(\mathbf{y}|\mathbf{x}_t)}_{\text{likelihood score}}. \tag{5}$$

The prior score is provided by the pretrained model via Eq. 2. Since only the likelihood $p(\mathbf{y}|\mathbf{x}_0)$ is given through the observation model, the central difficulty lies in approximating the intractable likelihood score $\nabla_{\mathbf{x}_t} \log p_t(\mathbf{y}|\mathbf{x}_t)$. Following Peng et al. (2024), existing methods generally fall into two categories.

**Direct likelihood approximation.** These methods directly approximate the likelihood score by marginalizing over a tractable surrogate for the denoising posterior $p(\mathbf{x}_0|\mathbf{x}_t)$. DPS (Chung et al., 2023) replaces $p(\mathbf{x}_0|\mathbf{x}_t)$ with a Dirac delta $\delta(\mathbf{x}_0 - \hat{\mathbf{x}}_0)$; ΠGDM (Song et al., 2023) uses an isotropic Gaussian $\mathcal{N}(\mathbf{x}_0 \, ; \, \hat{\mathbf{x}}_0, r_t^2 \mathbf{I})$, while TMPD (Boys et al., 2023) relaxes this assumption to a diagonal Gaussian. DSG (Yang et al., 2024) generalizes this perspective by framing the likelihood score through a spherical Gaussian surrogate, broadening the class of tractable guidance functions beyond the standard isotropic approximation.

**Proximal estimation.** These methods avoid direct likelihood approximation, and rather use the unconditional Tweedie estimate $\hat{\mathbf{x}}_0$ to construct a proximal problem in $\mathbf{x}_0$-space and compute an approximation to the conditional posterior mean $\mathbf{x}_0^* \approx \mathbb{E}[\mathbf{x}_0|\mathbf{x}_t, \mathbf{y}]$ by solving the optimization problem $\mathbf{x}_0^* = \arg\min_{\mathbf{x}_0} \big(\mathcal{L}(\mathbf{x}_0) + \lambda \mathcal{R}(\mathbf{x}_0)\big)$, with $\mathcal{L}$ being some data fidelity objective, $\mathcal{R}$ some regularization term, and $\lambda$ a weighting scalar. These problems can also be formulated within a Bayesian Maximum A-Posteriori (MAP) estimation approach, such that the proximal problem takes the form:

$$\mathbf{x}_0^* = \arg\max_{\mathbf{x}_0} \log p(\mathbf{x}_0 \mid \mathbf{x}_t, \mathbf{y}) = \arg\max_{\mathbf{x}_0} \big(\log p(\mathbf{x}_0 \mid \mathbf{x}_t) + \log p(\mathbf{y} \mid \mathbf{x}_0)\big). \tag{6}$$

While $\mathbf{x}_0^*$ is the MAP estimator, in linear settings where $\mathcal{A}(\mathbf{x}) = \mathbf{A}\mathbf{x}$ and under specific Gaussian assumptions, it coincides with the conditional mean $\mathbb{E}[\mathbf{x}_0 \mid \mathbf{x}_t, \mathbf{y}]$ (proof in Section A.2). It is important to note this equivalence is a specific property of the common linear-Gaussian assumption, rather than a general property of diffusion priors. For nonlinear operators, $\mathbf{x}_0^*$ serves as a tractable approximation of the mean, which is then substituted into the sampling path. Methods in this category include DDNM (Wang et al., 2023), DiffPIR (Zhu et al., 2023) (as interpreted by Peng et al. (2024)), DDPG (Garber and Tirer, 2024), and DAPS (Zhang et al., 2025), which samples from the distribution $p(\mathbf{x}_0 \mid \mathbf{x}_t, \mathbf{y})$ using Markov Chain Monte Carlo algorithms rather than estimating its mean.

**Comparison.** Direct methods offer a principled derivation but are computationally expensive, as they require backpropagation through the denoiser $\hat{\mathbf{x}}_0(\mathbf{x}_t, t)$ to compute gradients with respect to $\mathbf{x}_t$. Furthermore, as we discuss in this work, the Jacobian of a trained denoiser inherently reflects training-induced non-idealities, which can impact performance when used directly in zero-shot settings. In contrast, Proximal estimation approaches bypass this need but require heuristics to perform on par with direct methods that leverage the knowledge encoded in the learned denoiser. Both approaches are similar in their requirement of an assumption on the prior distribution of $\mathbf{x}_0$, via the regularization term $\mathcal{R}$ or via $p(\mathbf{x}_0 \mid \mathbf{x}_t)$.

## 3 Jacobian-Aware Posterior Sampling

This section presents the theoretical foundation of **J**acobian-**A**ware **P**osterior **S**ampling (JAPS). We bridge direct and proximal likelihood solvers (3.1), analyze the non-idealities of trained Jacobians 3.2, and propose a projection-based guidance mechanism (3.3). Finally, we reformulate DDIM for conditional sampling (3.4) and extend JAPS to nonlinear operators (3.5).

### 3.1 Bridging Direct and Proximal Likelihood Solvers

We begin by stating the following proposition:

**Proposition 3.1.** *The likelihood score admits the representation*

$$\nabla_{\mathbf{x}_t} \log p(\mathbf{y}|\mathbf{x}_t) = \frac{\sqrt{\bar{\alpha}_t}}{1 - \bar{\alpha}_t}\Big(\mathbb{E}[\mathbf{x}_0|\mathbf{x}_t, \mathbf{y}] - \mathbb{E}[\mathbf{x}_0|\mathbf{x}_t]\Big). \tag{7}$$

See the proof in Section A.1. This fundamental result suggests that when treating any proximal estimator from Eq. 6 as an approximation of $\mathbb{E}[\mathbf{x}_0|\mathbf{x}_t, \mathbf{y}]$, it can be used to form a likelihood approximated surrogate as in the direct approaches. However, because proximal problems are solved in $\mathbf{x}_0$-space, their corresponding surrogate does not acquire the denoiser's Jacobian factor $\mathbf{J}_t := (\partial \hat{\mathbf{x}}_0 / \partial \mathbf{x}_t)^\top$ – which arises naturally via the chain rule in direct methods to map the $\mathbf{x}_0$-space solution to the noisy intermediate state $\mathbf{x}_t$. In this work, we are interested in exploring the impact of $\mathbf{J}_t$ on posterior samplers.

Next, we formulate the settings to build a coupled pair of approximations, a direct and a proximal surrogate, that only differ by the presence of the Jacobian. Under the Gaussian isotropic assumption $p(\mathbf{x}_0 \mid \mathbf{x}_t) \sim \mathcal{N}(\mathbf{x}_0 \,;\, \hat{\mathbf{x}}_0, r_t^2 \mathbf{I})$, the proximal problem is quadratic thus the minimum can be approximately achieved via a single-step Gauss-Newton descent. With the further assumption that the measurement operator $\mathbf{A}$ is linear, it is guaranteed that $\arg\max_{\mathbf{x}_0} \log p(\mathbf{x}_0 \mid \mathbf{x}_t \,,\, \mathbf{y}) = \mathbb{E}[\mathbf{x}_0 \mid \mathbf{x}_t, \mathbf{y}]$ (proof in Section A.2). We can thus form the likelihood surrogate, which we denote by $\mathbf{v}_t$, using Proposition 3.1, similar to a regularized back-projection guidance (Garber and Tirer, 2024; Tirer and Giryes, 2018). In the direct approach, applying the same assumptions yields the solution presented in $\Pi$GDM (Song et al., 2023), which we denote by $\mathbf{u}_t$. Specifically,

$$\mathbf{v}_t := \frac{\sqrt{\bar{\alpha}_t}}{1-\bar{\alpha}_t} \mathbf{A}^\top (\mathbf{A}\mathbf{A}^\top + \frac{\sigma_y^2}{r_t^2}\mathbf{I})^{-1}(\mathbf{y} - \mathbf{A}\hat{\mathbf{x}}_0), \tag{8}$$

$$\mathbf{u}_t := \frac{1}{r_t^2} \mathbf{J}_t \mathbf{A}^\top (\mathbf{A}\mathbf{A}^\top + \frac{\sigma_y^2}{r_t^2}\mathbf{I})^{-1}(\mathbf{y} - \mathbf{A}\hat{\mathbf{x}}_0). \tag{9}$$

For detailed derivations see Sections A.3 and A.4.

In these settings, we get that $\mathbf{u}_t = \frac{1-\bar{\alpha}_t}{\sqrt{\bar{\alpha}_t}} \frac{1}{r_t^2} \mathbf{J}_t \mathbf{v}_t$. In order to tightly couple these direct and proximal approximations, we choose $r_t^2 = \frac{1-\bar{\alpha}_t}{\sqrt{\bar{\alpha}_t}}$ and obtain the pair $\mathbf{u}_t, \mathbf{v}_t$, such that $\mathbf{u}_t = \mathbf{J}_t \mathbf{v}_t$. Under this parameterization, we obtain a direct and a proximal likelihood surrogate that differ only by the presence of the Jacobian, both derived using the same set of assumptions. We discuss the choice of $r_t^2$ in Section B.1.

*Remark* 3.2. For many linear operators $\mathbf{A}$, the operator $\left(\mathbf{A}\mathbf{A}^\top + \frac{\sigma_y^2}{r_t^2}\mathbf{I}\right)^{-1}$ can be implemented efficiently: exactly via SVD (for small dense $\mathbf{A}$), via FFT diagonalization for circulant models (e.g., in super-resolution and deblurring), or via Conjugate Gradients in the general case.

### 3.1.1 Probabilistic Interpretation

**Proposition 3.3.** *Let $\hat{\mathbf{x}}_0(\mathbf{x}_t) = \mathbb{E}[\mathbf{x}_0|\mathbf{x}_t]$ be the ideal MMSE denoiser. Its Jacobian matrix with respect to the noisy input $\mathbf{x}_t$, defined as $\mathbf{J}_t = \nabla_{\mathbf{x}_t}\hat{\mathbf{x}}_0(\mathbf{x}_t)$, relates to the posterior covariance $\mathrm{Cov}(\mathbf{x}_0|\mathbf{x}_t)$ according to:*

$$\mathbf{J}_t = \frac{\sqrt{\bar{\alpha}_t}}{1 - \bar{\alpha}_t} \mathrm{Cov}(\mathbf{x}_0|\mathbf{x}_t). \tag{10}$$

Proof in Section A.5. Since proximal estimation approaches treat $\hat{\mathbf{x}}_0$ as a fixed-point, the application of the Jacobian factor in direct likelihood approximations can be viewed as *preconditioning* the proximal gradients according to the uncertainty curvature applied via the Jacobian, up to the scheduled constants.

### 3.2 Empirical Analysis of Jacobian Deviations

In practice, $\mathbb{E}[\mathbf{x}_0 \mid \mathbf{x}_t]$ is estimated using a neural network (NN) trained on a finite set of samples. Furthermore, these NNs are usually trained for denoising score matching (Meng et al., 2021; Song and Ermon, 2019), such that the Jacobian – a higher-order quantity – is not necessarily matched. These inevitably cause deviations of $\mathbf{J}_t$ from the ideal Jacobian, which we find harmful for posterior sampling.

Since $\mathbf{J}_t \propto \mathrm{Cov}[\mathbf{x}_0 \mid \mathbf{x}_t]$ for the MMSE denoiser, it is expected that an approximate Jacobian will at least be symmetric and Positive Semi-Definite (PSD). These properties are also required for $\mathbf{J}_t$ to act as a valid preconditioner, a role it implicitly plays in direct methods, as discussed in Section 3.1.1. We show that trained denoisers may violate these properties. To evaluate the non-ideality of trained Jacobians, we measure the minimum eigenvalue to prove PSD violation, and quantify the symmetry error. We compute both measures using an ImageNet-256 denoiser, pretrained by Dhariwal and Nichol (2021). To assess the distance from ideality, we construct a synthetic dataset of a Gaussian Mixture Model

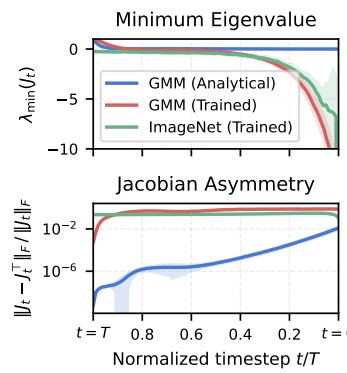

Figure 2: PSD and symmetry violations of trained Jacobians. The plots show mean curves with STDs over 1024/50 samples per time-step for the GMM/ImageNet curves.

(GMM), such that the ground truth distribution is known, and we can analytically compute the prior score $\nabla_{\mathbf{x}_t} \log p(\mathbf{x}_t)$, as well as the Jacobian of the MMSE denoiser and its properties. We also train a toy diffusion model to obtain an estimator for $\hat{\mathbf{x}}_0$, and show the difference between its Jacobian properties and those of an analytically derived Jacobian. Figure 2 shows that the minimal eigenvalue of a trained denoiser is *negative*, indicating it is not PSD, and that it significantly deviates from symmetry. Further details on the eigenvalues and symmetry measurements, and on the synthetic environment are given in Sections B.3 and B.4.

In this work, our main insight is that by constructing the direct and proximal likelihood approximations together, as shown in Section 3.1, we are able to mitigate harmful impact of the non-ideality of the Jacobian, while still leveraging the encoded knowledge within it, without extra computational cost.

### 3.3 The Proposed Guidance

Let us begin by constructing $\mathbf{u}_t$ and $\mathbf{v}_t$ as detailed in Section 3.1. We are interested in constructing an adaptive, enhanced likelihood surrogate $\mathbf{g}_t^*$ that leverages the knowledge within the Jacobian but is less sensitive to deviations resulting from non-idealities. Since $\mathbf{v}_t$ is already a decent approximation, it is natural to find the vector $\mathbf{g}_t \in \text{span}\{\mathbf{u}_t\}$ that is closest to $\mathbf{v}_t$. In other words, we optimize a constant $c_t$ for $\mathbf{g}_t = c_t \mathbf{u}_t$:

$$c_t^* = \arg\min_{c_t} \ \|c_t \mathbf{u}_t - \mathbf{v}_t\|_2^2 = \frac{\langle \mathbf{v}_t, \mathbf{u}_t \rangle}{\langle \mathbf{u}_t, \mathbf{u}_t \rangle} \quad \Longrightarrow \quad \mathbf{g}_t^* = \frac{\langle \mathbf{v}_t, \mathbf{u}_t \rangle}{\langle \mathbf{u}_t, \mathbf{u}_t \rangle} \mathbf{u}_t = \mathcal{P}_{\mathbf{u}_t}(\mathbf{v}_t). \tag{11}$$

Here, $\mathcal{P}_{\mathbf{u}_t}(\mathbf{v}_t)$ denotes the orthogonal projection of $\mathbf{v}_t$ onto $\mathbf{u}_t$. We discuss the rationale for this projection rule in Section B.5. Notice that while in general $\mathbf{u}_t$ and $\mathbf{v}_t$ may be computed differently, or conceptually even come from a decoupled pair of proximal and direct likelihood approximations, one can obtain both with no extra computational cost, relative to obtaining the direct solution $\mathbf{u}_t$ alone. By choosing $\mathbf{u}_t$ and $\mathbf{v}_t$ as in Eq. 9 and Eq. 8, $\mathbf{v}_t$ can be extracted from the same computational graph as $\mathbf{u}_t$ for free. Our method, therefore, suggests projecting the proximal solution $\mathbf{v}_t$ onto the direct, Jacobian-preconditioned solution $\mathbf{u}_t$, thus minimizing non-ideal Jacobian-related artifacts. Throughout the paper, we refer to our method as **J**acobian-**A**ware **P**osterior **S**ampling (JAPS).

### 3.4 Reformulating DDIM for Conditional Settings

Substituting Eq. 3 into the unconditional DDIM update Eq. 4, we can write a *Markovian DDIM update* as:

$$\mathbf{x}_{t-1} = \frac{1}{\sqrt{\alpha_t}} \mathbf{x}_t - \underbrace{\left( \frac{\sqrt{1 - \bar{\alpha}_t}}{\sqrt{\alpha_t}} - \sqrt{1 - \bar{\alpha}_{t-1} - \sigma_t^2} \right)}_{\gamma_t} \boldsymbol{\epsilon}_\theta(\mathbf{x}_t, t) + \sigma_t \, \boldsymbol{\epsilon}_t =: \text{DDIM}(\mathbf{x}_t), \tag{12}$$

where $\boldsymbol{\epsilon}_t \sim \mathcal{N}(\mathbf{0}, \mathbf{I})$, and $\gamma_t$ collects the time-dependent coefficients. Building on the conditional score identity $\mathbb{E}[\boldsymbol{\epsilon}|\mathbf{x}_t, \mathbf{y}] = -\sqrt{1 - \bar{\alpha}_t} \, \nabla_{\mathbf{x}_t} \log p_t(\mathbf{x}_t|\mathbf{y})$ (Peng et al., 2024), we define the *posterior noise estimator* $\tilde{\boldsymbol{\epsilon}}_\theta(\mathbf{x}_t, t, \mathbf{y})$ to relate directly to the posterior score. Using Eq. 5 and Eq. 2, we obtain:

$$\tilde{\boldsymbol{\epsilon}}_\theta(\mathbf{x}_t, t, \mathbf{y}) := \boldsymbol{\epsilon}_\theta(\mathbf{x}_t, t) - \lambda_t \sqrt{1 - \bar{\alpha}_t} \, \mathbf{g}(\mathbf{y}, \mathbf{x}_t), \tag{13}$$

where $\mathbf{g}(\mathbf{y}, \mathbf{x}_t)$ is any tractable estimator of the likelihood-score term $\nabla_{\mathbf{x}_t} \log p_t(\mathbf{y} \mid \mathbf{x}_t)$. Plugging the posterior noise Eq. 13 into the Markovian form Eq. 12 in lieu of $\boldsymbol{\epsilon}_\theta$ yields the final *conditional DDIM step*:

$$\mathbf{x}_{t-1} = \text{DDIM}(\mathbf{x}_t) + \gamma_t \, \lambda_t \, \sqrt{1 - \bar{\alpha}_t} \, \mathbf{g}(\mathbf{y}, \mathbf{x}_t). \tag{14}$$

This formulation makes explicit that posterior information affects the sample through the **time-step coefficient** $\gamma_t$. In contrast, most direct methods add the likelihood surrogate directly to the update without accounting for this factor (Chung et al., 2023; Song et al., 2023), while proximal methods often modify the noise estimate heuristically (Garber and Tirer, 2024; Zhu et al., 2023).

### 3.5 Extension For Nonlinear Measurement Operators

For a nonlinear forward operator $\mathcal{A}(\cdot)$, even under the isotropic Gaussian assumption for $p(\mathbf{x}_0 \mid \mathbf{x}_t)$, the resulting posterior $p(\mathbf{x}_0 \mid \mathbf{x}_t, \mathbf{y})$ is no longer Gaussian, causing the MAP estimate to deviate from the true conditional mean $\mathbb{E}[\mathbf{x}_0 \mid \mathbf{x}_t, \mathbf{y}]$. To restore tractability, we locally linearize $\mathcal{A}(\cdot)$ around the unconditional MMSE estimate $\hat{\mathbf{x}}_0 = \mathbb{E}[\mathbf{x}_0 \mid \mathbf{x}_t]$:

$$\mathcal{A}(\mathbf{x}_0) = \mathcal{A}(\hat{\mathbf{x}}_0) + \mathbf{J}_{\mathcal{A}}(\mathbf{x}_0 - \hat{\mathbf{x}}_0) + \mathcal{O}\big(\|\mathbf{H}_{\mathcal{A}}\|_2 \|\mathbf{x}_0 - \hat{\mathbf{x}}_0\|_2^2\big), \tag{15}$$

where $\mathbf{J}_{\mathcal{A}}$ and $\mathbf{H}_{\mathcal{A}}$ are the Jacobian and Hessian of $\mathcal{A}$ evaluated at $\hat{\mathbf{x}}_0$. For weakly nonlinear operators, the curvature penalty $\|\mathbf{H}_{\mathcal{A}}\|_2$ is negligible. Omitting these higher-order terms approximately restores the equivalence between the MAP estimate and the conditional mean.

Substituting the linearized operator into Eq. 6 yields the closed-form MAP update:

$$\mathbf{v}_t^{\mathrm{NL}} := \frac{\sqrt{\bar{\alpha}_t}}{1 - \bar{\alpha}_t} \mathbf{J}_{\mathcal{A}}^\top \big(\mathbf{J}_{\mathcal{A}} \mathbf{J}_{\mathcal{A}}^\top + \frac{\sigma_y^2}{r_t^2} \mathbf{I}\big)^{-1} \big(\mathbf{y} - \mathcal{A}(\hat{\mathbf{x}}_0)\big), \tag{16}$$

and correspondingly, $\mathbf{u}_t^{\mathrm{NL}} = \mathbf{J}_t \mathbf{v}_t^{\mathrm{NL}}$. For brevity, we omit the superscript $(\cdot)^{\mathrm{NL}}$ hereafter. In practice, explicitly constructing the full Jacobian matrix is intractable for high-dimensional data. Instead, the requisite vector-Jacobian products (VJPs) and Jacobian-vector products (JVPs) are efficiently computed via standard auto-differentiation, while the linear system defined by $(\mathbf{J}_{\mathcal{A}} \mathbf{J}_{\mathcal{A}}^\top + \frac{\sigma_y^2}{r_t^2} \mathbf{I})^{-1}$ is solved iteratively using the Conjugate Gradient (CG) method.

## 4 Experimental Results

**Tasks and datasets.** We test our method on the CelebA-HQ and ImageNet-256 validation sets, with backbone denoisers trained by Lugmayr et al. (2022) and Dhariwal and Nichol (2021), respectively. Our evaluation is performed by conducting several key image restoration tasks, used also in previous works: (i) *Super-resolution* ×4 with a bicubic downsampling kernel; (ii) *Gaussian deblurring* with a 5×5 Gaussian kernel (standard deviation 10); (iii) *Motion deblurring* with randomized 61×61 kernels of intensity 0.5, generated using the public implementation[1]; (iv) *Inpainting*, with masks of both a $128 \times 128$ box, or 90% random pixels (Box/Random Inpainting, respectively). Unless noted otherwise, we add zero-mean i.i.d. Gaussian noise with $\sigma_y = 0.05$, conventionally expressed in $[0,1]$ intensity units. Since we normalize images to the range $[-1,1]$, we accordingly multiply the noise level by a factor of 2.

**Baselines.** We compare against DAPS (Zhang et al., 2025), DiffPIR (Zhu et al., 2023), DDPG (Garber and Tirer, 2024), DPS (Chung et al., 2023) ΠGDM (Song et al., 2023), and DSG (Yang et al., 2024). The first 3 are referred to as proximal-typed solutions, while the last 3 are direct-typed works. We report the PSNR, SSIM, LPIPS (AlexNet variant) and FID metrics averaged or computed over 1K samples per dataset.

**Sampling details.** We use a DDIM sampler with $\eta = 1$ (i.e., DDPM-equivalent) and 100 diffusion steps, which amount to 100 NFEs. Unless noted otherwise, all baselines also use 100 NFEs. Exceptions are DPS and DAPS, which require 1,000 NFEs, and DSG, which likewise uses 1,000 steps for ImageNet-256.

**Visualizations.** Results from the experiments in Sections 4.1 and 4.2 are presented in Section E.

### 4.1 Linear Inverse Problems

Table 1 compares JAPS with representative posterior samplers built on diffusion priors. Across both datasets, JAPS delivers state-of-the-art perceptual quality, attaining the best or second-best LPIPS in all settings. At the same time, it maintains strong PSNR, incurring only a modest distortion cost, consistent with the perception–distortion trade-off (Blau and Michaeli, 2018). Notably, several PSNR-oriented baselines

---

[1]https://github.com/LeviBorodenko/motionblur

| | Method | Super Resolution | Gaussian Blur | Motion Blur | Random Inpainting | Box Inpainting |
|---|---|---|---|---|---|---|
| **CelebA-HQ** | DDPG | **29.40** / **0.838** / 0.086 / 40.95 | **30.42** / **0.853** / 0.054 / **35.94** | **29.01** / **0.827** / 0.066 / **37.22** | 25.63 / 0.780 / 0.146 / 54.59 | 25.98 / 0.852 / 0.073 / **30.11** |
| | DiffPIR | 26.58 / 0.763 / 0.081 / **38.78** | 28.92 / 0.813 / 0.064 / 37.32 | 27.34 / 0.783 / 0.078 / 39.14 | 26.74 / 0.808 / 0.138 / 52.45 | 25.88 / 0.859 / 0.096 / 46.27 |
| | DAPS | 28.12 / 0.781 / 0.075 / 28.63 | 28.34 / 0.756 / 0.091 / 32.66 | 28.42 / 0.794 / 0.072 / 26.36 | 26.65 / 0.753 / 0.120 / 38.14 | 24.14 / 0.723 / 0.165 / 33.74 |
| | DPS | 27.56 / 0.777 / 0.072 / 40.01 | 28.28 / 0.790 / 0.065 / 38.66 | 26.28 / 0.749 / 0.083 / 40.75 | 26.77 / 0.778 / 0.085 / 42.41 | 26.79 / 0.857 / 0.064 / 35.05 |
| | DSG | 27.57 / 0.780 / 0.072 / 40.77 | 30.29 / 0.845 / 0.051 / 36.05 | 27.57 / 0.790 / 0.079 / 38.48 | **28.39** / **0.828** / 0.079 / 43.56 | 25.47 / 0.855 / 0.068 / 36.54 |
| | IIGDM | 27.23 / 0.767 / 0.078 / 40.83 | 27.67 / 0.778 / 0.087 / 41.41 | 26.15 / 0.741 / 0.104 / 41.98 | 28.16 / 0.827 / 0.072 / **40.76** | 27.11 / **0.875** / 0.050 / 32.55 |
| | JAPS | 28.39 / 0.806 / **0.070** / 39.76 | 30.25 / 0.845 / **0.050** / 36.28 | 28.70 / 0.814 / **0.061** / 37.58 | 27.90 / 0.805 / **0.069** / 42.61 | **27.58** / 0.874 / **0.046** / 32.26 |
| **ImageNet** | DDPG | **25.55** / **0.722** / 0.302 / 68.34 | **27.74** / **0.793** / 0.170 / 37.15 | **25.90** / **0.735** / 0.210 / 42.08 | 21.85 / 0.602 / 0.395 / 146.84 | 20.98 / 0.749 / 0.206 / 61.34 |
| | DiffPIR | 23.53 / 0.643 / 0.331 / 64.35 | 24.56 / 0.645 / 0.192 / 38.86 | 24.13 / 0.665 / 0.334 / 70.57 | 22.27 / 0.617 / 0.376 / 133.02 | **21.07** / 0.761 / 0.244 / 85.88 |
| | DAPS | 23.60 / 0.626 / 0.422 / 77.95 | 21.74 / 0.504 / 0.331 / 124.94 | 25.14 / 0.696 / 0.240 / 51.09 | 22.80 / 0.595 / 0.309 / 81.09 | 20.26 / 0.675 / 0.248 / 57.67 |
| | DPS | 24.10 / 0.655 / 0.242 / 46.28 | 23.62 / 0.629 / 0.262 / 49.61 | 20.09 / 0.503 / 0.365 / 68.78 | 23.05 / 0.646 / 0.263 / 53.73 | 18.97 / 0.750 / 0.207 / 52.70 |
| | DSG | 24.33 / 0.669 / 0.203 / 39.89 | 26.69 / 0.744 / 0.153 / 30.19 | 24.33 / 0.667 / 0.197 / 41.23 | 21.91 / 0.576 / 0.236 / 54.88 | 19.18 / 0.769 / 0.151 / 40.13 |
| | IIGDM | 22.85 / 0.595 / 0.268 / 57.04 | 22.83 / 0.607 / 0.227 / 48.30 | 20.97 / 0.528 / 0.292 / 58.49 | **23.74** / 0.690 / 0.231 / 55.30 | 19.60 / **0.788** / 0.152 / 43.50 |
| | JAPS | 24.55 / 0.687 / **0.195** / 40.24 | 27.16 / 0.770 / **0.149** / 32.69 | 25.31 / 0.713 / **0.184** / 38.94 | 23.71 / **0.694** / **0.168** / 48.75 | 19.44 / 0.785 / **0.143** / **37.91** |

Table 1: Performance metrics on CelebA-HQ and ImageNet datasets. Cells show PSNR ↑ / SSIM ↑ / LPIPS ↓ / FID ↓. Best results are in bold, and second best are underlined among the methods using standard configurations. Gray-texted methods are evaluated using 1000 NFEs, and are excluded from the ranking.

significantly sacrifice LPIPS, especially on noisy tasks, resulting in visibly blurrier reconstructions (e.g., DDPG on SR×4, or DSG on Random Inpainting). In contrast, JAPS remains competitive on *both* metrics across tasks and datasets. These results are also visualized in Figure 1b, which averages the PSNR and LPIPS of each method across these linear tasks, demonstrating that, on average, our method excels in both perceptual and distortion metrics. Section C.1 reviews the empirical distribution of our results.

## 4.2 Nonlinear Inverse Problems

We test our method for nonlinear deblurring and high dynamic range correction as representative nonlinear tasks. Following Chung et al. (2023); Zhang et al. (2025), which are not naturally restricted to linear inverse problems, we adopt the neural blur model of Tran et al. (2021) with measurement noise $\sigma_y = 0.05$, and compare against them. Additionally, we evaluate high dynamic range correction with a factor of 2, also with measurement noise $\sigma_y = 0.05$, and compare to Zhang et al. (2025). In both cases we test on 100 CelebA-HQ and ImageNet-256 images. For a complete comparison, we also evaluate IIGDM (Song et al., 2023) with our nonlinear extension derived in Section 3.5. We report PSNR, SSIM and LPIPS. The results in Table 2 show that our nonlinear extension consistently performs best among compared methods, despite being an approximation.

| Method | ImageNet-256 | CelebA-HQ |
|---|---|---|
| | *Nonlinear Blur* | |
| DPS | 19.12 / 0.517 / 0.363 | 23.31 / 0.702 / 0.133 |
| DAPS | 19.98 / 0.364 / 0.586 | 24.75 / 0.667 / 0.144 |
| IIGDM | 21.87 / 0.598 / 0.345 | 25.52 / 0.754 / 0.109 |
| JAPS | **23.64** / **0.677** / **0.213** | **26.40** / **0.769** / **0.097** |
| | *High Dynamic Range* | |
| DAPS | 24.55 / 0.822 / 0.110 | 25.52 / 0.810 / 0.084 |
| IIGDM | 11.11 / 0.434 / 0.522 | 22.56 / 0.794 / 0.124 |
| JAPS | **24.70** / **0.841** / **0.099** | **26.74** / **0.848** / **0.067** |

Table 2: Comparison of nonlinear tasks across methods and datasets, measured by PSNR (↑) / SSIM (↑) / LPIPS (↓).

## 4.3 Ablation Studies

In this section, we analyze various aspects in our method.

**Analyzing our Likelihood Score Surrogate** In Section 3.3, we propose to orthogonally project a proximal likelihood score surrogate $\mathbf{v}_t$ onto a direct one $\mathbf{u}_t$ (e.g. the terms in Eq. 8 and Eq. 9). To validate that our solution in Eq. 11 indeed performs best, we vary our choice for the likelihood score surrogate $\mathbf{g}(\mathbf{y}, \mathbf{x}_t)$ from Eq. 14 by plugging each of the options $\mathbf{g}(\mathbf{y}, \mathbf{x}_t) \in \{\mathbf{u}_t, \mathbf{v}_t, \mathcal{P}_{\mathbf{v}_t}(\mathbf{u}_t), \mathcal{P}_{\mathbf{u}_t}(\mathbf{v}_t), (\mathbf{u}_t + \mathbf{v}_t)/2\}$. For a fair comparison, we sweep over different values $\lambda_t \in \{0.5, 1, 2, 3, 4, 5, 6, 7, 8\}$, and run 3 representative tasks on a subset of 30 ImageNet-256 images with a fixed seed. Figure 3 summarizes the results, and shows that our choice of $\mathbf{g}(\mathbf{y}, \mathbf{x}_t) = \mathcal{P}_{\mathbf{u}_t}(\mathbf{v}_t)$ attains the best performance across all tasks upon proper tuning.

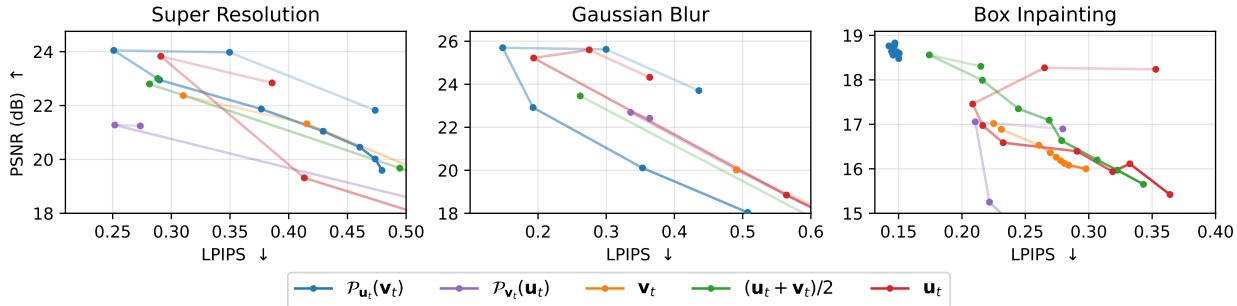

Figure 3: **Ablation study of guidance surrogates.** We compare the sampling scheme in Eq. 14 using various candidates for $\mathbf{g}(\mathbf{y}, \mathbf{x}_t) \in \{\mathbf{u}_t, \mathbf{v}_t, \mathcal{P}_{\mathbf{v}_t}(\mathbf{u}_t), \mathcal{P}_{\mathbf{u}_t}(\mathbf{v}_t), (\mathbf{u}_t + \mathbf{v}_t)/2\}$. The curves illustrate the perception–distortion trade-off (LPIPS vs. PSNR) on ImageNet-256, where each point corresponds to a different guidance scale $\lambda_t \in \{0.5, 1, 2, 3, 4, 5, 6, 7, 8\}$.

**Experiment on the Synthetic GMM Dataset.** Using the synthetic GMM settings from Section 3.2 and Section B.4, we analytically compute the ground-truth prior score. Given a degradation operator $\mathbf{A}$, we can also compute the likelihood score, the posterior score, and the full posterior distribution. We leverage this to evaluate and compare JAPS to its individual components, $\mathbf{u}$ and $\mathbf{v}$. In Section C.3, we conduct extensive experiments across multiple degradation operators, demonstrating the advantages of JAPS over the direct use of either the proximal or the direct likelihood surrogates. Notably, when using a perfect denoiser (derived from the analytical score), $\mathbf{u}_t$ generally performs best, empirically re-validating the value of the information encoded in the Jacobian. However, when using a trained model, the non-ideal properties of the Jacobian degrade performance–an effect that JAPS effectively mitigates.

**Additional Experiments.** We further evaluate JAPS under increasing measurement noise levels and varying degrees of stochasticity in the diffusion sampling process. Our results show that JAPS attenuates the unavoidable degradation in reconstruction quality at high noise levels and is substantially less sensitive to the stochasticity setting of the diffusion process. Additional details are provided in Section C.4.

## 5 Conclusion

We introduced **JAPS**, a principled approach for posterior sampling that bridges direct and proximal likelihood approximations. By leveraging the denoiser's Jacobian while mitigating its training-induced non-idealities, JAPS achieves a competitive balance between perceptual quality and distortion across diverse linear and nonlinear tasks.

**Limitations.** A primary limitation is the current reliance on pixel-domain diffusion. Additionally, while highly effective in practice, our theoretical derivations rely on the widely adopted assumptions of an isotropic Gaussian surrogate for the denoising posterior and locally linear measurements, which do not transfer to latent models. Extending JAPS to latent diffusion models (Rombach et al., 2022) is non-trivial, as posterior sampling becomes entangled with the decoder when measurements are defined in the image domain (Rout et al., 2023; Song et al., 2024). Furthermore, while we focus on diffusion models, current state-of-the-art generative modeling is increasingly shifting toward flow-based models (Lipman et al., 2023; Liu et al., 2022). Adapting our insights regarding the Jacobian of the trained backbone to such architectures remains a promising direction for future research.

**Outlook.** To the best of our knowledge, this is the first work to analyze the impact of Jacobian non-ideality in zero-shot posterior sampling settings. We believe JAPS provides a mathematically grounded foundation for more reliable inverse problem solving without task-specific tuning. Future work should further explore methods to explicitly adjust or regularize the desired properties of the Jacobian to better suit specific downstream tasks, facilitating more robust posterior guidance in both pixel and latent generative models.

## Acknowledgment

TT was supported by ISF grant No. 1940/23 and MOST grant No. 0007091. RG was supported by a grant from the Tel Aviv University Center for AI and Data Science (TAD).

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

# A    Proofs and Derivations

## A.1    Proof of Proposition 3.1

*Proof.* By Bayes' rule, $p(\mathbf{y} \mid \mathbf{x}_t) = p(\mathbf{x}_t \mid \mathbf{y})\, p(\mathbf{y}) \,/\, p(\mathbf{x}_t)$. Taking the gradient of the logarithm with respect to $\mathbf{x}_t$,

$$\nabla_{\mathbf{x}_t} \log p(\mathbf{y} \mid \mathbf{x}_t) = \nabla_{\mathbf{x}_t} \log p(\mathbf{x}_t \mid \mathbf{y}) - \nabla_{\mathbf{x}_t} \log p(\mathbf{x}_t). \tag{17}$$

The rightmost term is the score, which relates to the mean $\mathbb{E}[\mathbf{x}_0 \mid \mathbf{x}_t]$ via Tweedie's Formula (Efron, 2011):

$$\nabla_{\mathbf{x}_t} \log p(\mathbf{x}_t) = \frac{-\mathbf{x}_t + \sqrt{\bar{\alpha}_t}\, \mathbb{E}[\mathbf{x}_0 \mid \mathbf{x}_t]}{1 - \bar{\alpha}_t}. \tag{18}$$

Now, the assumption $\mathbf{x}_t \perp\!\!\!\perp \mathbf{y} \mid \mathbf{x}_0$ implies $p(\mathbf{x}_t \mid \mathbf{x}_0, \mathbf{y}) = p(\mathbf{x}_t \mid \mathbf{x}_0)$, so the class-conditional marginal factorises as

$$p(\mathbf{x}_t \mid \mathbf{y}) = \int p(\mathbf{x}_t \mid \mathbf{x}_0)\, p(\mathbf{x}_0 \mid \mathbf{y})\, d\mathbf{x}_0.$$

$p(\mathbf{x}_0 \mid \mathbf{y})$ in place of $p(\mathbf{x}_0)$ in Tweedie's Formula, the posterior that appears is now $p(\mathbf{x}_0 \mid \mathbf{x}_t, \mathbf{y})$, giving

$$\nabla_{\mathbf{x}_t} \log p(\mathbf{x}_t \mid \mathbf{y}) = \frac{-\mathbf{x}_t + \sqrt{\bar{\alpha}_t}\, \mathbb{E}[\mathbf{x}_0 \mid \mathbf{x}_t, \mathbf{y}]}{1 - \bar{\alpha}_t}. \tag{19}$$

Substituting Eq. 18 and Eq. 19 into Eq. 17, the $\mathbf{x}_t$ terms cancel, yielding

$$\nabla_{\mathbf{x}_t} \log p(\mathbf{y} \mid \mathbf{x}_t) = \frac{\sqrt{\bar{\alpha}_t}}{1 - \bar{\alpha}_t}\Big(\mathbb{E}[\mathbf{x}_0 \mid \mathbf{x}_t, \mathbf{y}] - \mathbb{E}[\mathbf{x}_0 \mid \mathbf{x}_t]\Big). \qquad \square$$

**Remark.** This identity is exact and requires no Gaussian approximation on $p(\mathbf{x}_0 \mid \mathbf{x}_t)$. It decomposes the intractable likelihood score into the difference of two posterior means, scaled by the signal-to-noise geometry of the forward process.

## A.2    Proof of Mean-Mode Coincidence for a Linear-Gaussian Model

**Proposition A.1.** *Let* $\mathbf{y} = \mathbf{A}\mathbf{x}_0 + \boldsymbol{\eta}$, $\boldsymbol{\eta} \sim \mathcal{N}(\mathbf{0}, \sigma_y^2 \mathbf{I})$, $q(\mathbf{x}_t \mid \mathbf{x}_0) = \mathcal{N}(\sqrt{\bar{\alpha}_t}\mathbf{x}_0, (1 - \bar{\alpha}_t)\mathbf{I})$ *the forward diffusion kernel, with* $\mathbf{x}_t \perp\!\!\!\perp \mathbf{y} \mid \mathbf{x}_0$. *If* $p(\mathbf{x}_0 \mid \mathbf{x}_t)$ *is Gaussian, then* $\arg\max_{\mathbf{x}_0} \log p(\mathbf{x}_0 \mid \mathbf{x}_t, \mathbf{y}) = \mathbb{E}[\mathbf{x}_0 \mid \mathbf{x}_t, \mathbf{y}]$.

*Proof.* By Bayes' rule and the conditional independence assumption,

$$\log p(\mathbf{x}_0 \mid \mathbf{x}_t, \mathbf{y}) = \underbrace{\log p(\mathbf{y} \mid \mathbf{x}_0)}_{\text{likelihood}} + \underbrace{\log p(\mathbf{x}_0 \mid \mathbf{x}_t)}_{\text{prior}} + \text{const.}$$

The prior is Gaussian by assumption. The likelihood $p(\mathbf{y} \mid \mathbf{x}_0) = \mathcal{N}(\mathbf{A}\mathbf{x}_0, \sigma_y^2 \mathbf{I})$ is quadratic in $\mathbf{x}_0$ because $\mathbf{A}$ is linear. The sum of two quadratics is quadratic, so $p(\mathbf{x}_0 \mid \mathbf{x}_t, \mathbf{y})$ is Gaussian. Since the mode and mean of a Gaussian coincide, the MAP estimate equals the posterior mean $\mathbb{E}[\mathbf{x}_0 \mid \mathbf{x}_t, \mathbf{y}]$. $\qquad \square$

## A.3    Derivation of Eq. 8

Let $\Phi(\mathbf{x}_0) = -\log p(\mathbf{x}_0 | \mathbf{x}_t, \mathbf{y})$. Under the isotropic Gaussian assumption on $p(\mathbf{x}_0 \mid \mathbf{x}_t)$ and the linearity of $\mathbf{A}$, $\Phi(\mathbf{x}_0)$ gets the form:

$$\Phi(\mathbf{x}_0) = \frac{1}{2\sigma_y^2} \|\mathbf{y} - \mathbf{A}\mathbf{x}_0\|_2^2 + \frac{1}{2r_t^2} \|\mathbf{x}_0 - \hat{\mathbf{x}}_0\|_2^2 + C. \tag{20}$$

Moreover, under these assumptions, $\mathbf{x}_0^* = \arg\min_{\mathbf{x}_0} \Phi(\mathbf{x}_0) = \mathbb{E}[\mathbf{x}_0 \mid \mathbf{x}_t, \mathbf{y}]$.

Since $\Phi$ is quadratic, its minima can be achieved using a single step from any initial point $\mathbf{x}_p$, in the Newton descent direction, defined by $\mathbf{d} = -\mathbf{H}^{-1}\mathbf{g}$, where $\mathbf{g} := \nabla_{\mathbf{x}_0}\Phi(\mathbf{x}_0)|_{\mathbf{x}_0=\mathbf{x}_p}$ and $\mathbf{H} := \nabla^2_{\mathbf{x}_0}\Phi(\mathbf{x}_0)|_{\mathbf{x}_0=\mathbf{x}_p}$, the gradient and the Hessian at the point $\mathbf{x}_p$.

Taking $\mathbf{x}_p = \hat{\mathbf{x}}_0$, we get $\mathbf{g} = -\sigma_y^{-2}\mathbf{A}^\top(\mathbf{y} - \mathbf{A}\hat{\mathbf{x}}_0)$, and $\mathbf{H} = \sigma_y^{-2}\mathbf{A}^\top\mathbf{A} + r_t^{-2}\mathbf{I}$, thus,

$$\mathbf{d} = \left(\mathbf{A}^\top\mathbf{A} + \tfrac{\sigma_y^2}{r_t^2}\mathbf{I}\right)^{-1}\mathbf{A}^\top(\mathbf{y} - \mathbf{A}\hat{\mathbf{x}}_0) \tag{21}$$

For any $\mathbf{A} \in \mathbb{R}^{n \times m}$, recall the "push-through" identity,

$$\left(\mathbf{A}^\top\mathbf{A} + \lambda\mathbf{I}_n\right)^{-1}\mathbf{A}^\top = \mathbf{A}^\top\left(\mathbf{A}\mathbf{A}^\top + \lambda\mathbf{I}_m\right)^{-1} \qquad \forall\,\lambda > 0, \tag{22}$$

Using Eq. 22, we get $\mathbf{d} = \mathbf{A}^\top\left(\mathbf{A}\mathbf{A}^\top + \tfrac{\sigma_y^2}{r_t^2}\mathbf{I}\right)^{-1}(\mathbf{y} - \mathbf{A}\hat{\mathbf{x}}_0)$

Hence, since the optimum $\mathbf{x}_0^*$ is achieved with a single step descent $\mathbf{d}$ from $\hat{\mathbf{x}}_0$, $\mathbf{x}_0^* - \hat{\mathbf{x}}_0 = \mathbf{d}$. Plugging this into Prop. 3.1, we get the expression for $\mathbf{v}_t$ in Eq. 8.

### A.4 Derivation of Eq. 9

Directly from Song et al. (2023),

$$\nabla_{\mathbf{x}_t}\log p(\mathbf{y} \mid \mathbf{x}_t) = \left(\frac{\partial\hat{\mathbf{x}}_0}{\partial\mathbf{x}_t}\right)^\top\mathbf{A}^\top\left(r_t^2\mathbf{A}\mathbf{A}^\top + \sigma_y^2\mathbf{I}\right)^{-1}(\mathbf{y} - \mathbf{A}\hat{\mathbf{x}}_0). \tag{23}$$

Taking $r_t^2$ outside of $\left(r_t^2\mathbf{A}\mathbf{A}^\top + \sigma_y^2\mathbf{I}\right)^{-1}$, we get the expression of $\mathbf{u}_t$ in Eq. 9.

### A.5 Proof of Proposition 3.3

*Proof.* By the definition of the conditional expectation, the Jacobian $\mathbf{J}_t = \nabla_{\mathbf{x}_t}\mathbb{E}[\mathbf{x}_0|\mathbf{x}_t]$ is given by:

$$\mathbf{J}_t = \nabla_{\mathbf{x}_t}\int\mathbf{x}_0 p(\mathbf{x}_0|\mathbf{x}_t)d\mathbf{x}_0 \tag{24}$$

Assuming the data distribution $p(\mathbf{x}_0)$ has finite moments, the Gaussian forward transition $p(\mathbf{x}_t|\mathbf{x}_0)$ guarantees that the posterior $p(\mathbf{x}_0|\mathbf{x}_t)$ and its derivatives with respect to $\mathbf{x}_t$ decay exponentially. Under these regularity conditions, the Leibniz integral rule permits the interchange of the gradient and the integral:

$$\mathbf{J}_t = \int\mathbf{x}_0\nabla_{\mathbf{x}_t}p(\mathbf{x}_0|\mathbf{x}_t)^\top d\mathbf{x}_0 \tag{25}$$

Applying the log-derivative identity, $\nabla_{\mathbf{x}_t}p(\mathbf{x}_0|\mathbf{x}_t) = p(\mathbf{x}_0|\mathbf{x}_t)\nabla_{\mathbf{x}_t}\log p(\mathbf{x}_0|\mathbf{x}_t)$, yielding:

$$\mathbf{J}_t = \int\mathbf{x}_0\left(\nabla_{\mathbf{x}_t}\log p(\mathbf{x}_0|\mathbf{x}_t)\right)^\top p(\mathbf{x}_0|\mathbf{x}_t)d\mathbf{x}_0 \tag{26}$$

Using Bayes' theorem, the log-posterior is $\log p(\mathbf{x}_0|\mathbf{x}_t) = \log p(\mathbf{x}_t|\mathbf{x}_0) + \log p(\mathbf{x}_0) - \log p(\mathbf{x}_t)$. Differentiating with respect to $\mathbf{x}_t$ gives:

$$\nabla_{\mathbf{x}_t}\log p(\mathbf{x}_0|\mathbf{x}_t) = \nabla_{\mathbf{x}_t}\log p(\mathbf{x}_t|\mathbf{x}_0) - \nabla_{\mathbf{x}_t}\log p(\mathbf{x}_t) \tag{27}$$

Given the forward noising process $\mathbf{x}_t|\mathbf{x}_0 \sim \mathcal{N}(\sqrt{\bar{\alpha}_t}\mathbf{x}_0, (1 - \bar{\alpha}_t)\mathbf{I})$, the gradient of the log-likelihood is:

$$\nabla_{\mathbf{x}_t}\log p(\mathbf{x}_t|\mathbf{x}_0) = -\frac{\mathbf{x}_t - \sqrt{\bar{\alpha}_t}\mathbf{x}_0}{1 - \bar{\alpha}_t} \tag{28}$$

By Tweedie's formula, the gradient of the log-marginal distribution is related to the MMSE denoiser via:

$$\nabla_{\mathbf{x}_t}\log p(\mathbf{x}_t) = \frac{\sqrt{\bar{\alpha}_t}\mathbb{E}[\mathbf{x}_0|\mathbf{x}_t] - \mathbf{x}_t}{1 - \bar{\alpha}_t} \tag{29}$$

Substituting these into Equation 27:

$$\nabla_{\mathbf{x}_t} \log p(\mathbf{x}_0|\mathbf{x}_t) = -\frac{\mathbf{x}_t - \sqrt{\bar{\alpha}_t}\mathbf{x}_0}{1 - \bar{\alpha}_t} - \frac{\sqrt{\bar{\alpha}_t}\mathbb{E}[\mathbf{x}_0|\mathbf{x}_t] - \mathbf{x}_t}{1 - \bar{\alpha}_t} \tag{30}$$

$$= \frac{\sqrt{\bar{\alpha}_t}}{1 - \bar{\alpha}_t}(\mathbf{x}_0 - \mathbb{E}[\mathbf{x}_0|\mathbf{x}_t]) \tag{31}$$

Substituting this result back into the integral for $\mathbf{J}_t$:

$$\mathbf{J}_t = \frac{\sqrt{\bar{\alpha}_t}}{1 - \bar{\alpha}_t} \int \mathbf{x}_0(\mathbf{x}_0 - \mathbb{E}[\mathbf{x}_0|\mathbf{x}_t])^\top p(\mathbf{x}_0|\mathbf{x}_t)d\mathbf{x}_0 \tag{32}$$

$$= \frac{\sqrt{\bar{\alpha}_t}}{1 - \bar{\alpha}_t}\mathbb{E}\left[\mathbf{x}_0(\mathbf{x}_0 - \mathbb{E}[\mathbf{x}_0|\mathbf{x}_t])^\top \big| \mathbf{x}_t\right] \tag{33}$$

$$= \frac{\sqrt{\bar{\alpha}_t}}{1 - \bar{\alpha}_t}\text{Cov}(\mathbf{x}_0|\mathbf{x}_t) \tag{34}$$

This completes the proof. □

## B   Additional Method Details

### B.1   The Choice of $r_t^2$

While Song et al. (2023) set $r_t^2 = 1 - \bar{\alpha}_t$ in ΠGDM, we propose $r_t^2 = (1 - \bar{\alpha}_t)/\sqrt{\bar{\alpha}_t}$. Both choices yield a decaying noise level along the reverse process, from $t = T$ toward the clean image at $t = 0$. However, under the widely adopted schedules in which $\text{Var}(\mathbf{x}_{t+1} \mid \mathbf{x}_t) = \text{Var}(\mathbf{x}_t \mid \mathbf{x}_{t-1})$ for every $t$, the original choice $r_t^2 = 1 - \bar{\alpha}_t$ remains nearly constant from $t = T$ to $t = T/2$ (see Figure 4). Combined with a fixed $\sigma_y$ throughout sampling, this results in an essentially unchanged prior-data trade-off over a large portion of the reverse process. Our choice, by contrast, exhibits a consistently steeper decay across the entire sampling trajectory, leading to a more gradual and effective transition from prior to data.

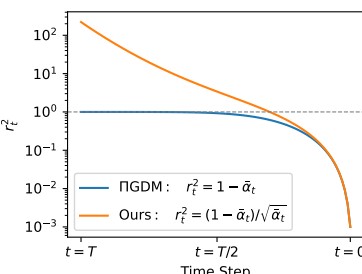

Figure 4: $r_t^2$ schedules across the diffusion process.

### B.2   Measurement Space Computations

In many inverse problems, $m < n$ (e.g. super-resolution). In such cases, it is preferable in terms of memory consumption to compute $\mathbf{u}_t$, $\mathbf{v}_t$ in the measurement space using Eq. 22. Our implementations indeed perform such computations in the measurement space, i.e:

$$\mathbf{v}_t = \frac{\sqrt{\bar{\alpha}_t}}{1 - \bar{\alpha}_t}\mathbf{A}^\top(\mathbf{A}\mathbf{A}^\top + \frac{\sigma_y^2}{r_t^2}\mathbf{I}_m)^{-1}(\mathbf{y} - \mathbf{A}\hat{\mathbf{x}}_0) \tag{35}$$

$$\mathbf{u}_t = \frac{1}{r_t^2}\left(\frac{\partial \hat{\mathbf{x}}_0}{\partial \mathbf{x}_t}\right)^\top\mathbf{A}^\top(\mathbf{A}\mathbf{A}^\top + \frac{\sigma_y^2}{r_t^2}\mathbf{I}_m)^{-1}(\mathbf{y} - \mathbf{A}\hat{\mathbf{x}}_0) \tag{36}$$

Furthermore, this measurement-space form is tightly connected to $\mathbf{A}^\dagger = \mathbf{A}^\top(\mathbf{A}\mathbf{A}^\top)$, the Moore-Penrose pseudo inverse of $\mathbf{A}$. In noiseless cases ($\sigma_y = 0$), the forms in Eq. 35 coincide with the Moore-Penrose pseudo inverse. In noisy cases, this form is a variant of the Moore-Penrose pseudo inverse, loaded with a regularization scalar $\eta_t := \frac{\sigma_y^2}{r_t^2}$.

### B.3   Jacobian Properties Estimation

For an ideal MMSE denoiser $D_\theta(\mathbf{x}_t, t)$, the Jacobian $\mathbf{J} = \partial D_\theta/\partial \mathbf{x}_t$ is symmetric and positive semi-definite (PSD), as it is proportional to the posterior covariance $\text{Cov}[\mathbf{x}_0 \mid \mathbf{x}_t]$. We empirically measure three properties

of $\mathbf{J}$ across timesteps to quantify deviations from this ideal. Because $\mathbf{J} \in \mathbb{R}^{d \times d}$ with $d = C \times H \times W$ is far too large to materialise, all estimates use only Jacobian–vector products $\mathbf{Jv}$ (forward-mode autodiff) and vector–Jacobian products $\mathbf{J}^\top \mathbf{v}$ (reverse-mode autodiff).

**Spectral norm $\sigma_{\max}(\mathbf{J})$.** The largest singular value is estimated via power iteration on $\mathbf{J}^\top \mathbf{J}$. Starting from a random unit vector $\mathbf{v}_0$, each iteration computes

$$\mathbf{u}_k = \mathbf{J}\,\mathbf{v}_k, \qquad \mathbf{z}_k = \mathbf{J}^\top \mathbf{u}_k, \qquad \sigma_{\max} \approx \|\mathbf{z}_k\|^{1/2}, \qquad \mathbf{v}_{k+1} = \mathbf{z}_k/\|\mathbf{z}_k\|. \tag{37}$$

We use $K{=}15$ iterations by default. This gives an upper bound on the Lipschitz constant of the denoiser at each noise level.

**Smallest eigenvalue of the symmetric part $\lambda_{\min}(\mathbf{S})$.** Let $\mathbf{S} = (\mathbf{J} + \mathbf{J}^\top)/2$ be the symmetric part of the Jacobian. We estimate its smallest eigenvalue via *shifted* power iteration on the matrix $(\mu\mathbf{I} - \mathbf{S})$, where the shift $\mu = \sigma_{\max} + 0.1$ ensures that the operator's largest eigenvalue corresponds to the smallest eigenvalue of $\mathbf{S}$. Each iteration computes

$$\mathbf{Sv}_k = \tfrac{1}{2}\big(\mathbf{Jv}_k + \mathbf{J}^\top \mathbf{v}_k\big), \qquad \mathbf{w}_k = \mu\,\mathbf{v}_k - \mathbf{Sv}_k, \qquad \mathbf{v}_{k+1} = \mathbf{w}_k/\|\mathbf{w}_k\|. \tag{38}$$

After $K{=}30$ iterations, the eigenvalue is recovered via the Rayleigh quotient $\lambda_{\min} \approx \mathbf{v}_K^\top \mathbf{S}\,\mathbf{v}_K$. Negative values of $\lambda_{\min}$ indicate that the Jacobian's symmetric part is not PSD at that input, violating the ideal denoiser assumption. We report both the mean $\lambda_{\min}$ and the fraction of samples with $\lambda_{\min} < 0$ at each timestep.

**Relative symmetry error.** We estimate the relative Frobenius-norm asymmetry $\|\mathbf{J} - \mathbf{J}^\top\|_F/\|\mathbf{J}\|_F$ using a Hutchinson-style stochastic trace estimator. For $M$ random unit probes $\{\mathbf{v}_m\}_{m=1}^M$, we accumulate

$$\|\mathbf{J} - \mathbf{J}^\top\|_F^2 \approx \sum_{m=1}^{M} \|(\mathbf{J} - \mathbf{J}^\top)\mathbf{v}_m\|^2 = \sum_{m=1}^{M} \|\mathbf{Jv}_m - \mathbf{J}^\top \mathbf{v}_m\|^2, \tag{39}$$

and similarly $\|\mathbf{J}\|_F^2 \approx \sum_m \|\mathbf{Jv}_m\|^2$. The ratio of their square roots gives the relative error ($0 =$ perfectly symmetric). We use $M{=}10$ probes by default.

**Protocol.** At each evaluated timestep $t$, we sample $N{=}50$ clean images $\mathbf{x}_0$ from the validation set and form $\mathbf{x}_t = \sqrt{\bar{\alpha}_t}\,\mathbf{x}_0 + \sqrt{1 - \bar{\alpha}_t}\,\boldsymbol{\epsilon}$ with $\boldsymbol{\epsilon} \sim \mathcal{N}(\mathbf{0}, \mathbf{I})$. All three quantities are computed per-sample and then aggregated (mean, std, min, max) per timestep. When the model predicts both the mean and variance (i.e. the output has $2C$ channels), only the first $C$ channels (the mean prediction) are used, keeping the Jacobian square.

### B.4 Synthetic GMM Environment

To enable controlled analysis of Jacobian properties, we construct a synthetic environment based on a Gaussian Mixture Model (GMM) in $\mathbb{R}^D$, where the ground-truth distribution and all derived quantities are available in closed form.

**GMM Construction.** We define a $K$-component GMM with density $p(\mathbf{x}_0) = \sum_{k=1}^{K} w_k \mathcal{N}(\mathbf{x}_0; \boldsymbol{\mu}_k, \boldsymbol{\Sigma}_k)$, where $w_k = 1/K$ are uniform weights. The component means $\{\boldsymbol{\mu}_k\}_{k=1}^K$ are placed on a hypersphere of radius $r{=}3$ by normalizing i.i.d. Gaussian random vectors. Each covariance matrix is constructed as $\boldsymbol{\Sigma}_k = \mathbf{Q}_k \operatorname{diag}(\boldsymbol{\lambda}_k) \mathbf{Q}_k^\top$, where $\mathbf{Q}_k$ is a random orthogonal matrix (via QR decomposition of a Gaussian random matrix) and eigenvalues $\lambda_{k,i}$ are drawn uniformly from $[0, 0.2]$. The main results use $D{=}256$ and $K{=}8$.

Under the forward diffusion process $\mathbf{x}_t = \sqrt{\bar{\alpha}_t}\,\mathbf{x}_0 + \sqrt{1 - \bar{\alpha}_t}\,\boldsymbol{\epsilon}$, the noisy marginal $p(\mathbf{x}_t)$ remains a GMM with evolved parameters:

$$p(\mathbf{x}_t) = \sum_{k=1}^{K} w_k \mathcal{N}\big(\mathbf{x}_t;\ \sqrt{\bar{\alpha}_t}\,\boldsymbol{\mu}_k,\ \bar{\alpha}_t \boldsymbol{\Sigma}_k + (1 - \bar{\alpha}_t)\mathbf{I}\big). \tag{40}$$

The prior score $\nabla_{\mathbf{x}_t} \log p(\mathbf{x}_t)$ is computed in closed form via the mixture gradient, and serves as the analytical denoiser through $\hat{\mathbf{x}}_0(\mathbf{x}_t) = \mathbb{E}[\mathbf{x}_0 \mid \mathbf{x}_t]$. To analyze the Jacobian $\mathbf{J}_t = \partial \hat{\mathbf{x}}_0 / \partial \mathbf{x}_t$, we compute the full $D \times D$ Jacobian matrix using automatic differentiation (`torch.autograd.functional.jacobian`) applied to the denoiser mapping $\mathbf{x}_t \mapsto \hat{\mathbf{x}}_0(\mathbf{x}_t)$. This procedure is identical for both the analytical MMSE denoiser and the trained neural network. Given the full Jacobian $\mathbf{J}_t$, we assess PSD violation via the minimum eigenvalue of its symmetric part $\lambda_{\min}\big((\mathbf{J}_t + \mathbf{J}_t^\top)/2\big)$, and asymmetry via the relative Frobenius norm $\|\mathbf{J}_t - \mathbf{J}_t^\top\|_F / \|\mathbf{J}_t\|_F$. Both quantities are computed per sample at each diffusion timestep. Figure 2 reports means and standard deviations of $\lambda_{\min}(\mathbf{J}_t)$ and $\|\mathbf{J}_t - \mathbf{J}_t^\top\|_F / \|\mathbf{J}_t\|_F$ over 1024 samples.

*Remark* B.1. Because the quadratic form $\mathbf{v}^\top \mathbf{J}_t \mathbf{v}$ is strictly equal to $\mathbf{v}^\top \left( \frac{\mathbf{J}_t + \mathbf{J}_t^\top}{2} \right) \mathbf{v}$ for any real vector $\mathbf{v}$, $\mathbf{J}_t$ is PSD if and only if its symmetric part is PSD. Thus, a negative minimum eigenvalue of the symmetric part is a definitive proof of PSD violation for the original matrix $\mathbf{J}_t$.

**Trained Diffusion Model.** We train a small MLP-based diffusion model on samples from the GMM. The architecture consists of a sinusoidal time embedding followed by an MLP with 256 hidden channels and 4 residual blocks, predicting the noise $\boldsymbol{\epsilon}$ via the standard denoising score matching objective $\mathbb{E}_{\mathbf{x}_0, \boldsymbol{\epsilon}, t} \big[ \|\boldsymbol{\epsilon} - \boldsymbol{\epsilon}_\theta(\mathbf{x}_t, t)\|^2 \big]$. The model is trained for 10,000 epochs using AdamW with learning rate $5 \times 10^{-4}$, weight decay $10^{-6}$, cosine annealing schedule, and batch size 1,024. We use a linear noise schedule with $T{=}1{,}000$ diffusion steps and sample with DDIM using 100 steps.

## B.5    Discussion on the Proposed Guidance in Eq. 11

As detailed in Section 3.3, we propose projecting the proximal likelihood score approximation $\mathbf{v}_t$ onto the 1D span of the direct solution $\mathbf{u}_t$. These surrogates are constructed such that $\mathbf{u}_t = \mathbf{J}_t \mathbf{v}_t$. Consequently, the resulting likelihood surrogate is defined as $\mathbf{g}_t^* = c_t^* \mathbf{u}_t$, where $c_t^* = \frac{\langle \mathbf{v}_t, \mathbf{u}_t \rangle}{\langle \mathbf{u}_t, \mathbf{u}_t \rangle}$ represents the optimal scalar for the orthogonal projection. We now examine several considerations regarding this formulation.

**Motivation for the 1D Constraint.** The restriction of $\mathbf{g}_t$ to the span of $\mathbf{u}_t$ is a principled choice to preserve the prior geometry encoded in the Jacobian while mitigating its numerical instabilities. Our synthetic analysis in Section C.3 confirms that with an ideal denoiser, the direct solution $\mathbf{u}_t$ provides a more accurate approximation of the true likelihood score than the proximal surrogate $\mathbf{v}_t$. Furthermore, ablation studies on trained backbones (Section C.2) indicate that solutions leaning toward the direct surrogate generally achieve superior perceptual results. While $\mathbf{u}_t$ captures essential local geometry of the data prior, it can be unreliable due to training non-idealities. We therefore utilize the proximal solution $\mathbf{v}_t$, which prioritizes measurement consistency, as a structural regularizer to calibrate the scaling of $\mathbf{u}_t$. A higher-dimensional optimization—for instance, seeking a matrix $\mathbf{W}$ to minimize $\|\mathbf{W}\mathbf{u}_t - \mathbf{v}_t\|_2^2$—would either collapse to the proximal estimate $\mathbf{v}_t$ (discarding the Jacobian information) or require computationally prohibitive operations involving $\mathbf{J}_t$. Consequently, the 1D projection serves as an efficient middle ground that preserves the Jacobian's directional guidance while enforcing a magnitude consistent with the measurement constraints. Future research may generalize this framework by considering generic functions $\mathbf{f}(\mathbf{u}_t, \mathbf{v}_t)$ as likelihood surrogates.

**Edge Cases.** We analyze the behavior of $c_t^*$ in two critical scenarios: $c_t^* \approx 0$: Assuming a bounded $\|\mathbf{u}_t\|_2^2$, this indicates that $\mathbf{u}_t$ and $\mathbf{v}_t$ are approximately orthogonal. In this case, JAPS adaptively suppresses the likelihood guidance ($\mathbf{g}_t^* \approx 0$), allowing the prior score to dictate the local sampling step. $c_t^* < 0$: Substituting $\mathbf{u}_t = \mathbf{J}_t \mathbf{v}_t$ into the numerator of $c_t^*$ implies that $\mathbf{v}_t^\top \mathbf{J}_t \mathbf{v}_t < 0$, which confirms that the symmetric part of $\mathbf{J}_t$ is not positive semi-definite (PSD). Here, JAPS reverses the direction dictated by $\mathbf{u}_t$ to align with the proximal estimate $\mathbf{v}_t$.

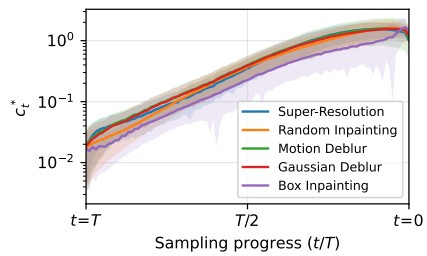

Figure 5: Evolution of $c_t^*$ (mean and standard deviation) over the sampling process.

Both scenarios identify mathematically inconsistent states arising from non-ideal backbones. However, Figure 5 displays the evolution of $c_t^*$, demonstrating that such states are atypical.

# C   Additional Results

## C.1   Empirical Distribution Visualization

To demonstrate the statistical significance of our results and quantify the distribution of the evaluation metrics, we provide violin plots summarizing the performance reported in Table 1. These plots display the mean values alongside kernel density estimates that represent the distribution of each metric. The results are presented in Figure 6, showing that across both datasets, JAPS exhibits improved performance compared to existing baselines without sacrificing stability in terms of standard deviation or worst-case outcomes.

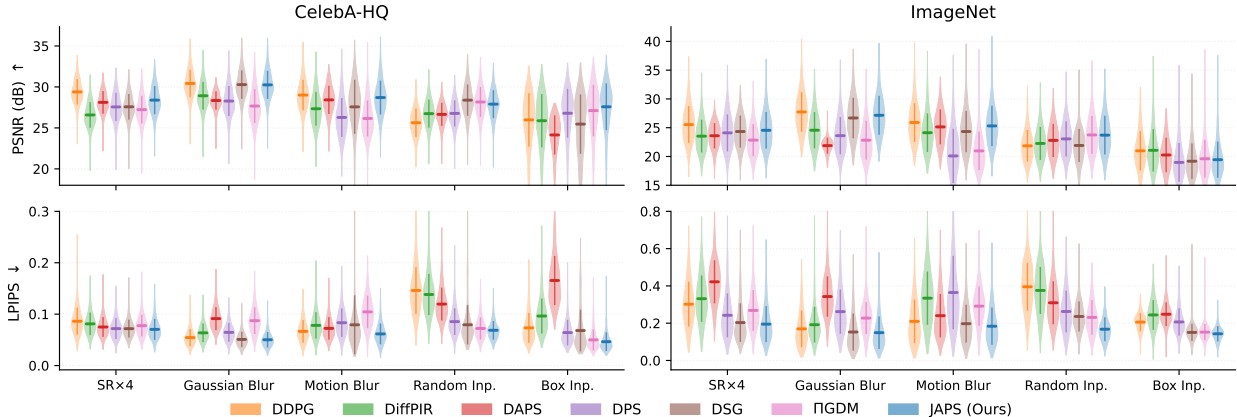

Figure 6: **Statistical distribution of reconstruction metrics.** Violin plots of the results from Table 1. For each task, the central marker indicates the mean, and the violin shape represents the density of results.

## C.2   Comparison to Convex Combination of Likelihood Surrogates

Our proposed likelihood surrogate from Eq. 11 preserves the direction of the direct likelihood solution $\mathbf{u}_t$ while inheriting the magnitude of the proximal solution $\mathbf{v}_t$. Given the potential harm of Jacobian non-idealities, an alternative approach would be to blend both magnitude and direction by constructing a convex combination of the two solutions; namely, for $\beta \in [0, 1]$, we define $\mathbf{g}_t = \beta \mathbf{u}_t + (1 - \beta)\mathbf{v}_t$.

In this section, we evaluate this approach across various values of $\beta$ and compare it to JAPS on the task of $\times 4$ super-resolution using 100 images from the ImageNet-256 dataset. For this experiment, we consistently set $\lambda_t = 1$ across all likelihood surrogates.

The results, presented in Figure 7, demonstrate that JAPS outperforms a simple convex combination of the surrogates used in its construction. While certain values of $\beta$ show slightly higher performance in terms of PSNR, this improvement does not translate to SSIM, which also measures distortion. In contrast, JAPS exhibits a significant improvement in LPIPS, confirming its superior perceptual fidelity.

## C.3   Experiments on the Synthetic GMM Dataset

To validate the efficacy of JAPS, defined as $\mathcal{P}_{\mathbf{u}_t}(\mathbf{v}_t)$, against its individual components $\mathbf{u}_t$ and $\mathbf{v}_t$, we conduct controlled experiments within the synthetic GMM environment detailed in Section B.4. This setup allows us to control the data prior, generate known measurement operators, and analytically compute the likelihood score $\nabla_{\mathbf{x}_t} \log p(\mathbf{y} \mid \mathbf{x}_t)$ and the posterior distribution $p(\mathbf{x}_t \mid \mathbf{y})$. We utilize a 256-dimensional GMM with 8 modes and randomized covariance matrices following the protocol in Section B.4.

**Measurement operators.**   We generate four types of measurement operators, enumerated I, II, III and IV, distinguished by their spectral decay profiles and output dimensions (see Figure 8). While I-III map the 256d input to a 32d measurement vector, IV maintains a $256 \times 256$ dimensionality. These operators are

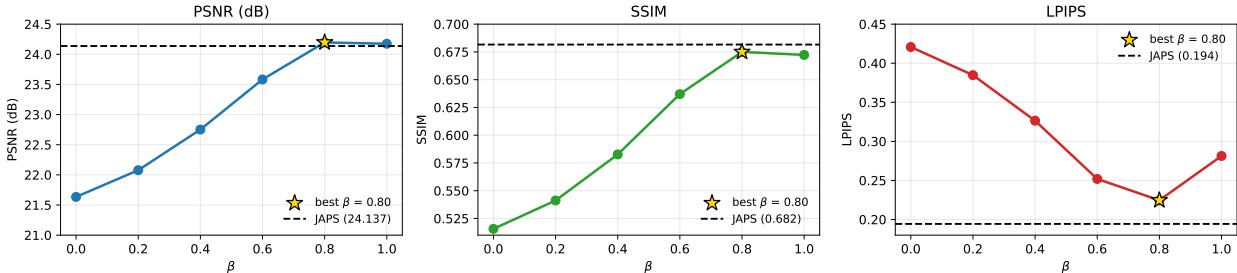

Figure 7: **Ablation Study:** Comparison of JAPS against a convex combination of surrogates, $\mathbf{g}_t = \beta\mathbf{u}_t + (1-\beta)\mathbf{v}_t$, incorporated into Eq. 14 with $\lambda_t = 1$.

constructed via SVD, where the spectral profile determines the singular values and the singular matrices are generated via QR decomposition. We generate 5 operators per type for thorough evaluation.

**Evaluation.** We compare JAPS against the direct use of $\mathbf{u}_t$ and $\mathbf{v}_t$ by sweeping over the scaling hyperparameter $\lambda_t$, as the presence or absence of the Jacobian significantly alters the update magnitude. For each task, we report two metrics at the optimal scale $\lambda_t$: (i) the mean $L_2$ error between the likelihood surrogate and the ground-truth likelihood score, and (ii) the Wasserstein-2 ($W_2$) distance between the sampled and ground-truth terminal posterior distributions.

Figure 9a illustrates that while the direct use of $\mathbf{u}_t$ is typically more stable and achieves the lowest $W_2$ error at the optimal scale for an analytical denoiser, its performance significantly degrades when using a trained denoiser. In the latter scenario,

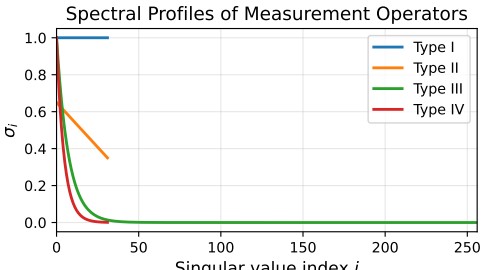

Figure 8: Spectral profile of the measurement operators.

JAPS enhances stability and yields improved $W_2$ errors. Figure 9b examines the $W_2$ error for each likelihood surrogate at its respective optimal scale, demonstrating that JAPS effectively mitigates the degradation in posterior distribution estimation inherent to trained denoisers. Notably, the use of $\mathbf{v}_t$ exhibits substantial instability across different measurement operator types. Finally, Figure 9c presents the $L_2$ error between each likelihood surrogate and the ground-truth likelihood score throughout the sampling process. Although direct use of $\mathbf{u}_t$ provides the best approximation of the ideal likelihood score for an analytical denoiser, it diverges when utilizing a trained denoiser; under these conditions, JAPS achieves the most accurate approximation.

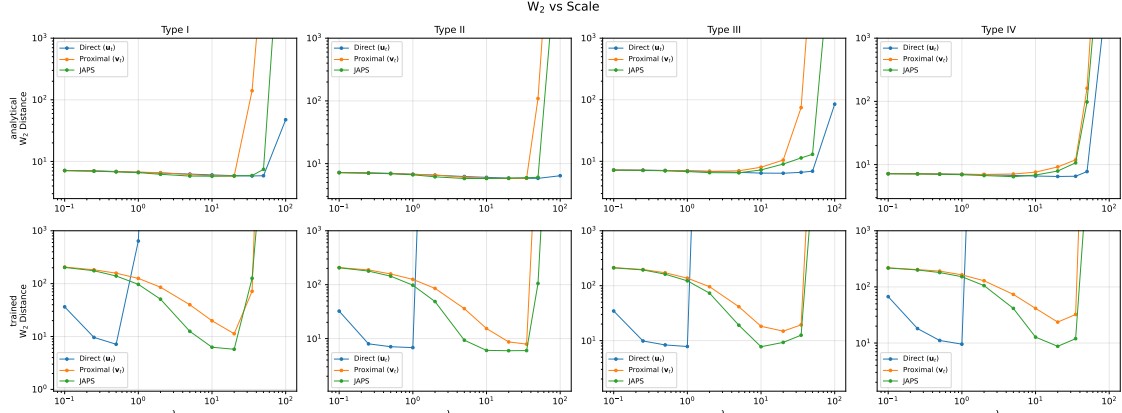

(a) $W_2$ **error vs.** $\lambda_t$. While $\mathbf{u}_t$ provides the most stable performance for analytical denoisers, this trend shifts for trained denoisers. JAPS introduces a correction that stabilizes the error and, in some cases, slightly outperforms the direct use of $\mathbf{u}_t$.

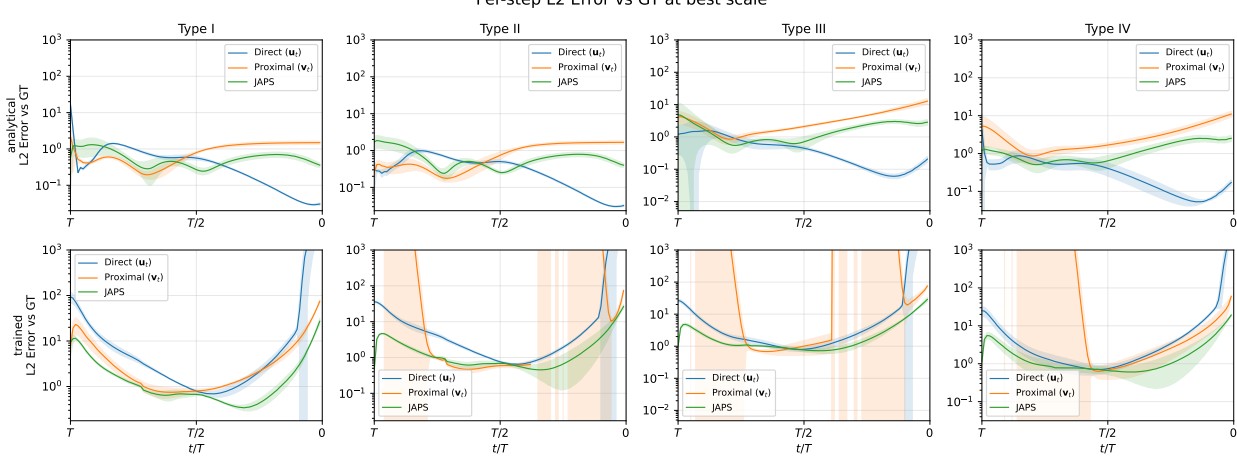

(b) $W_2$ **differences per method at optimal scale.** We inspect the mean $W_2$ error across operator types at each method's best scale, using standard deviation as a measure of stability. JAPS exhibits the smallest performance degradation when transitioning from analytical to trained denoisers.

(c) **Per-step $L_2$ error between approximated and ground-truth likelihood scores.** While $\mathbf{u}_t$ best approximates the true likelihood under ideal conditions (analytical score), JAPS provides the most accurate overall approximation when utilizing a trained neural network.

Figure 9: **Comparison of likelihood surrogates in the synthetic GMM environment.** We evaluate JAPS against its direct ($\mathbf{u}_t$) and proximal ($\mathbf{v}_t$) components. (a) Analysis of posterior sampling accuracy via $W_2$ distance across guidance scales. (b) Robustness across diverse operator types, highlighting that JAPS minimizes performance loss caused by denoiser non-ideality. (c) Direct measurement of score approximation accuracy, demonstrating that JAPS compensates for training-induced errors in the denoiser Jacobian.

## C.4 Additional Experiments

**Measurement noise level.** In many posterior sampling schemes, increasing the observation noise not only removes information but can also *leak* through the measurement-consistency gradients into the reconstruction. We evaluate SR×4 across increasing $\sigma_y$ and compare JAPS to ΠGDM using PSNR and LPIPS. We conduct this comparison against ΠGDM due to the structural resemblance of the likelihood surrogates, differing mainly in magnitude. Figure 10 shows that JAPS mitigates the inevitable degradation, with quality declining roughly linearly as the noise grows, whereas ΠGDM exhibits a markedly sharper drop.

**Stochasticity.** Many posterior samplers set $\eta = 1$ (see Section 2.2), allowing fresh noise to mitigate artifacts that arise when enforcing consistency with noisy observations. We evaluate JAPS on SR×4 across a range of $\eta$ values and compare against ΠGDM. Figure 11 shows that, whereas ΠGDM benefits primarily from highly stochastic updates (large $\eta$), JAPS exhibits markedly weaker dependence on $\eta$, maintaining similar performance across stochasticity levels.

**Direct Comparison to ΠGDM.** Beyond the likelihood approximation, Song et al. (2023) introduce a time-decaying multiplicative step size equal to $(1 - \bar{\alpha}_t)$. While this choice performs well at 100 sampling

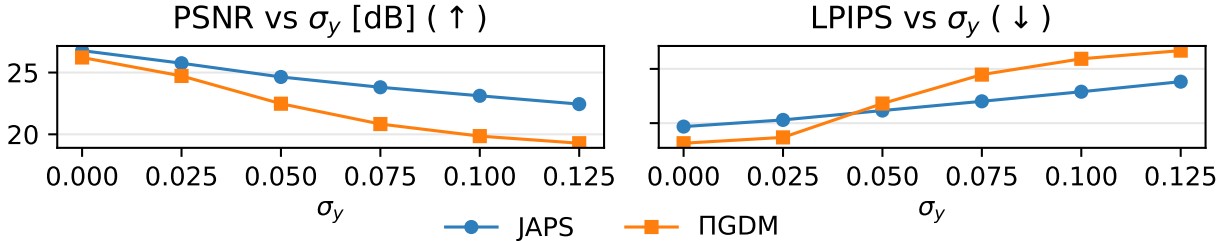

Figure 10: **Response to increasing measurement noise level.** JAPS shows enhanced robustness to increased noise compared to ΠGDM.

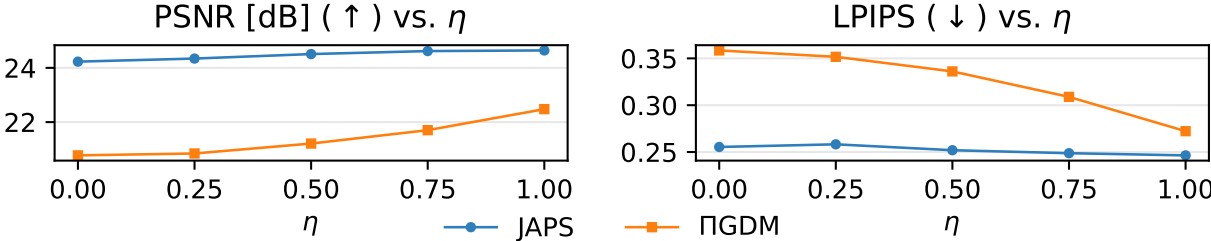

Figure 11: **Sensitivity to stochasticity ($\eta$).** PSNR/LPIPS for SR×4 as a function of $\eta$. JAPS maintains similar performance across stochasticity levels, whereas ΠGDM is more sensitive to the amount of injected noise.

steps, it does not account for the diffusion schedule's discretization (i.e., changing the number of steps). Consequently—and counter-intuitively—*increasing* the number of steps degrades performance in both PSNR and LPIPS, rather than improving it, as was also observed by Mardani et al. (2023). By solving SR×4 with $\sigma_y = 0.05$ on 100 samples from the ImageNet-256 validation set, we show that our method explicitly incorporates step spacing through $\gamma_t$, yielding a correctly scalable sampler (Figs. 12a and 12b).

# D   Implementation Details

## D.1   Hyperparameters

In this section, we detail the values of the hyperparameter $\lambda_t$ as used in our final experiments, and discuss their contribution. Table 3 details the values.

| Task | Bicub. SR×4 $\sigma_y = 0.05$ | Gauss.Deb. $\sigma_y = 0.05$ | Motion Deb. $\sigma_y = 0.05$ | Random Inp. $\sigma_y = 0.05$ | Box Inp. $\sigma_y = 0.05$ | Nonlinear Deb. $\sigma_y = 0.05$ | HDR $\sigma_y = 0.05$ |
|---|---|---|---|---|---|---|---|
| ImageNet | $4\sqrt{1 - \bar{\alpha}_t}$ | 1.8 | 1.8 | 6 | $19\sqrt{1 - \bar{\alpha}_t}$ | 4 | 4 |
| CelebA-HQ | 1 | 1.25 | 1.25 | 6 | $16\sqrt{1 - \bar{\alpha}_t}$ | $4\sqrt{1 - \bar{\alpha}_t}$ | 4 |

Table 3: Hyperparameter $\lambda_t$ of our method.

While these hyperparameters improve performance, tuning them is not the primary source of improvement in our method relative to others. To justify this claim, we show a direct comparison of JAPS with different hyperparameters and also of ΠGDM Song et al. (2023) using the same sweep. Specifically, we scale the likelihood score surrogate by a time-dependent constant in lieu of the hyperparameter $\lambda_t$ from Eq. 14. We note that for ΠGDM, the $\sqrt{1 - \bar{\alpha}_t}$ term is absent; we instead use their proposed adaptive weights and scale them to align both methods within the same sweep settings. When using the original ΠGDM scaling (e.g.,

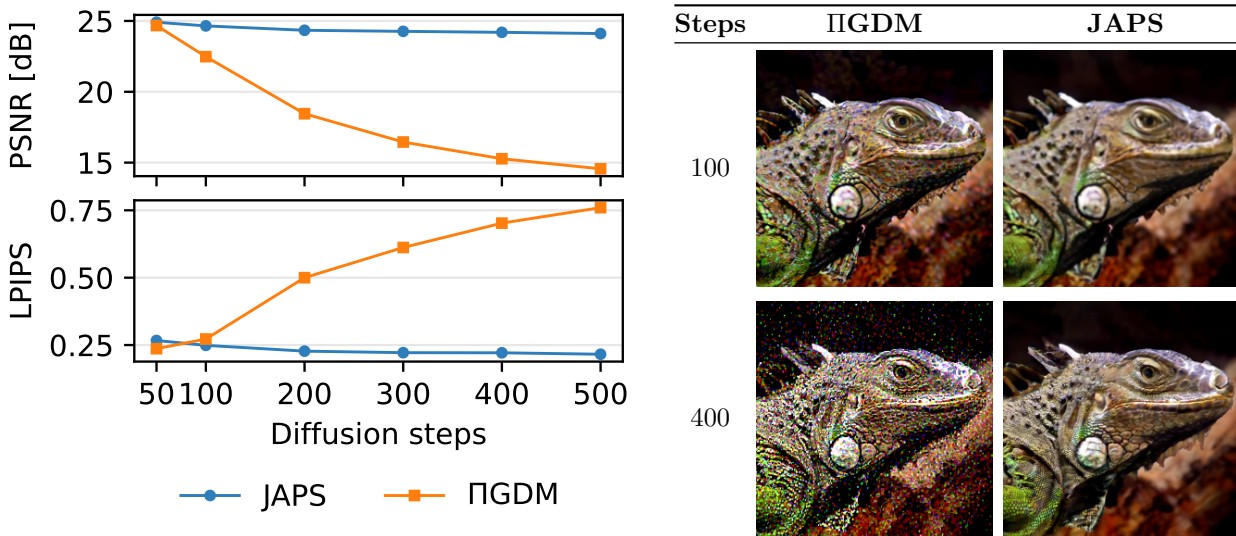

(a) Quantitative comparison. JAPS scales with the number of steps, while ΠGDM does not.

(b) Qualitative comparison. ΠGDM deteriorates at larger step counts, while JAPS remains stable.

Figure 12: Overall comparison of JAPS and ΠGDM across varying step counts. JAPS demonstrates consistent scaling both quantitatively (a) and qualitatively (b).

1), we find that the optimal choice of $\lambda_t$ for our method surpasses ΠGDM performance in both PSNR and LPIPS across almost all tasks for ImageNet-256. We note that a 0.5 scale for ΠGDM performs better than the default 1 scaling for most tasks, however, our method still either improves beyond this or stays on par. The full results are shown in Figure 13, indicating that the adaptiveness of the JAPS surrogate to the proximal solution cannot be replicated by simple time-dependent scaling.

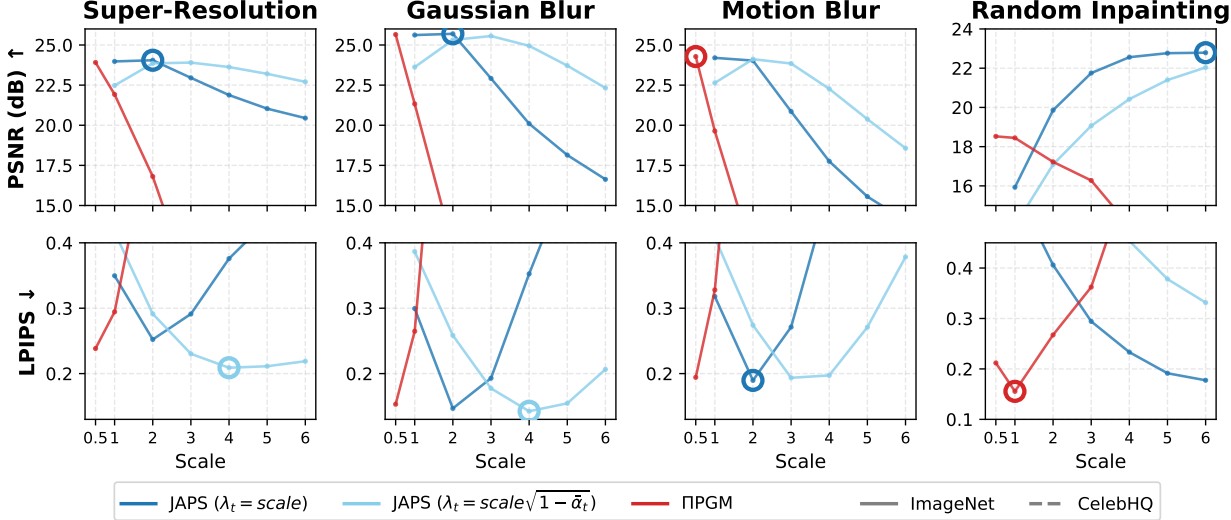

Figure 13: Hyperparameter sweep for ImageNet-256. The best scores per task are circled.

## D.2 Computational Efficiency and Timings

In this section, we present inference time measurements for JAPS and the reproduced baselines. Notably, JAPS incurs only marginal computational overhead relative to ΠGDM. This empirically validates our asser-

| Task | Method | ImageNet | | CelebA-HQ | | Task | Method | ImageNet | | CelebA-HQ | |
|---|---|---|---|---|---|---|---|---|---|---|---|
| | | NFE | Time (s) | NFE | Time (s) | | | NFE | Time (s) | NFE | Time (s) |
| SR ×4 | DDPG | 100 | $4.44 \pm 0.16$ | 100 | $2.41 \pm 0.80$ | Rand. Inp. | DDPG | 100 | $4.31 \pm 0.16$ | 100 | $2.32 \pm 0.80$ |
| | DiffPIR | 100 | $4.41 \pm 0.17$ | 100 | $2.40 \pm 0.80$ | | DiffPIR | 100 | $4.31 \pm 0.16$ | 100 | $2.31 \pm 0.82$ |
| | DAPS | 1000 | $62.07 \pm 0.42$ | 1000 | $41.99 \pm 1.25$ | | DAPS | 1000 | $46.68 \pm 0.44$ | 1000 | $27.18 \pm 0.98$ |
| | DPS | 1000 | $84.07 \pm 0.47$ | 1000 | $46.97 \pm 1.17$ | | DPS | 1000 | $83.00 \pm 0.44$ | 1000 | $44.76 \pm 0.90$ |
| | DSG | 1000 | $47.05 \pm 0.42$ | 100 | $3.49 \pm 0.09$ | | DSG | 1000 | $50.74 \pm 0.38$ | 100 | $4.47 \pm 0.09$ |
| | ΠGDM | 100 | $8.43 \pm 0.33$ | 100 | $4.72 \pm 0.87$ | | ΠGDM | 100 | $8.34 \pm 0.33$ | 100 | $4.68 \pm 0.88$ |
| | JAPS | 100 | $8.42 \pm 0.34$ | 100 | $4.70 \pm 0.88$ | | JAPS | 100 | $8.31 \pm 0.34$ | 100 | $4.59 \pm 0.88$ |
| Gauss. Deb. | DDPG | 100 | $4.47 \pm 0.16$ | 100 | $2.38 \pm 0.80$ | Box Inp. | DDPG | 100 | $4.32 \pm 0.16$ | 100 | $2.28 \pm 0.81$ |
| | DiffPIR | 100 | $4.40 \pm 0.16$ | 100 | $2.39 \pm 0.80$ | | DiffPIR | 100 | $4.36 \pm 0.16$ | 100 | $2.28 \pm 0.80$ |
| | DAPS | 1000 | $59.35 \pm 0.58$ | 1000 | $39.78 \pm 1.36$ | | DAPS | 1000 | $45.90 \pm 0.23$ | 1000 | $26.93 \pm 0.82$ |
| | DPS | 1000 | $84.07 \pm 0.47$ | 1000 | $46.15 \pm 0.92$ | | DPS | 1000 | $82.50 \pm 0.43$ | 1000 | $44.65 \pm 0.88$ |
| | DSG | 1000 | $51.34 \pm 0.37$ | 100 | $4.60 \pm 0.09$ | | DSG | 1000 | $50.82 \pm 0.41$ | 100 | $4.50 \pm 0.11$ |
| | ΠGDM | 100 | $8.40 \pm 0.34$ | 100 | $4.67 \pm 0.88$ | | ΠGDM | 100 | $8.43 \pm 0.34$ | 100 | $4.68 \pm 0.87$ |
| | JAPS | 100 | $8.42 \pm 0.33$ | 100 | $4.66 \pm 0.88$ | | JAPS | 100 | $8.32 \pm 0.33$ | 100 | $4.67 \pm 1.12$ |
| Motion Deb. | DDPG | 100 | $4.34 \pm 0.15$ | 100 | $2.33 \pm 0.81$ | Nonlin. Deb. | DAPS | 4000 | $486.48 \pm 76.86$ | 4000 | $327.62 \pm 1.47$ |
| | DiffPIR | 100 | $4.38 \pm 0.16$ | 100 | $2.35 \pm 0.80$ | | DPS | 1000 | $89.83 \pm 0.36$ | 1000 | $52.21 \pm 0.85$ |
| | DAPS | 1001 | $48.45 \pm 0.33$ | 1000 | $28.48 \pm 0.90$ | | ΠGDM | 100 | $47.72 \pm 2.72$ | 100 | $46.19 \pm 1.70$ |
| | DPS | 1000 | $83.26 \pm 0.41$ | 1000 | $45.02 \pm 0.91$ | | JAPS | 100 | $60.33 \pm 1.50$ | 100 | $47.40 \pm 1.30$ |
| | DSG | 1000 | $50.83 \pm 0.42$ | 100 | $4.52 \pm 0.09$ | HDR | DAPS | 4000 | $167.58 \pm 1.22$ | 4000 | $86.77 \pm 0.90$ |
| | ΠGDM | 100 | $8.42 \pm 0.33$ | 100 | $4.67 \pm 0.88$ | | ΠGDM | 100 | $14.31 \pm 0.56$ | 100 | $4.97 \pm 0.92$ |
| | JAPS | 100 | $8.40 \pm 0.33$ | 100 | $4.64 \pm 0.89$ | | JAPS | 100 | $8.62 \pm 0.36$ | 100 | $4.98 \pm 0.89$ |

Table 4: NFEs and wall-clock time (seconds, mean $\pm$ std over 30 images) for each (task, method, dataset) combination. Hardware: NVIDIA L40S.

tion that the method requires no significant additional cost beyond the computation of the direct likelihood surrogate $\mathbf{u}_t$. Furthermore, although JAPS utilizes Conjugate Gradient iterations for nonlinear operators (see Section 3.5), alternative nonlinear-compatible methods typically require significantly more Neural Function Evaluations (NFEs). Consequently, JAPS demonstrates superior efficiency in terms of total wall-clock time. All benchmarks were performed using NVIDIA L40S GPUs.

## D.3 Reproducibility

### D.3.1 Baselines Details

We reproduce all baselines within a unified environment to ensure consistency across tasks, noise levels, and datasets. To verify the accuracy of our reproduction, we first replicated the results reported in the original papers for standard tasks. Once consistency was confirmed, we evaluated each method across the full suite of tasks and datasets considered in this work.

**DDPG.** We adopt the implementation details and hyperparameters reported by Garber and Tirer (2024). For inpainting tasks not covered in the original work, we conducted a grid search for each hyperparameter. The optimized values are summarized in Table 5.

Table 5: DDPG inpainting hyperparameters.

| Task | CelebA-HQ | ImageNet |
|---|---|---|
| Random Inpainting ($\sigma_y = 0.05$) | $\gamma = 3.0,\ \zeta = 0.9,\ \tilde{\eta} = 0.1,\ \mu_t = \mu_t^*$ | $\gamma = 9.0,\ \zeta = 0.9,\ \tilde{\eta} = 0.1,\ \mu_t = \mu_t^*$ |
| Box Inpainting ($\sigma_y = 0.05$) | $\gamma = 9.0,\ \zeta = 0.8,\ \tilde{\eta} = 0.9,\ \mu_t = \mu_t^*$ | $\gamma = 9.0,\ \zeta = 1.0,\ \tilde{\eta} = 0.7,\ \mu_t = \mu_t^*$ |

**DiffPIR.** We follow the implementation details from Zhu et al. (2023), including all reported hyperparameters. For our CelebA-HQ experiments, we utilize the hyperparameters specified by Zhu et al. (2023) for the FFHQ dataset.

**DAPS.** We follow the implementation in Zhang et al. (2025). In our framework, we scale $\sigma_y$ by a factor of 2 to account for the difference in image normalization (shifting from $[0, 1]$ to $[-1, 1]$). Consequently, our

reproduction of Zhang et al. (2025) effectively solves tasks with twice the original noise level. To compensate for this, we adjusted only the $\sigma_{\min}$ value of the annealing process from 0.1 to 0.2, while retaining all other original parameters.

**DPS.** We adopt the implementation details and hyperparameters from Chung et al. (2023) as reported.

**DSG.** We utilize the implementation details from Yang et al. (2024). For CelebA-HQ, we adopt the FFHQ hyperparameters. For motion deblurring, we employ the hyperparameters originally reported for Gaussian deblurring, as we found these tasks to be closely related in performance behavior.

**ΠGDM.** We adopt the implementation from Song et al. (2023), following the publicly released RED-diff codebase (Mardani et al., 2023)[2].

Our code will be released upon acceptance.

---

[2]`https://github.com/NVlabs/RED-diff/blob/master/algos/pgdm.py`

# E    Additional Visual Results

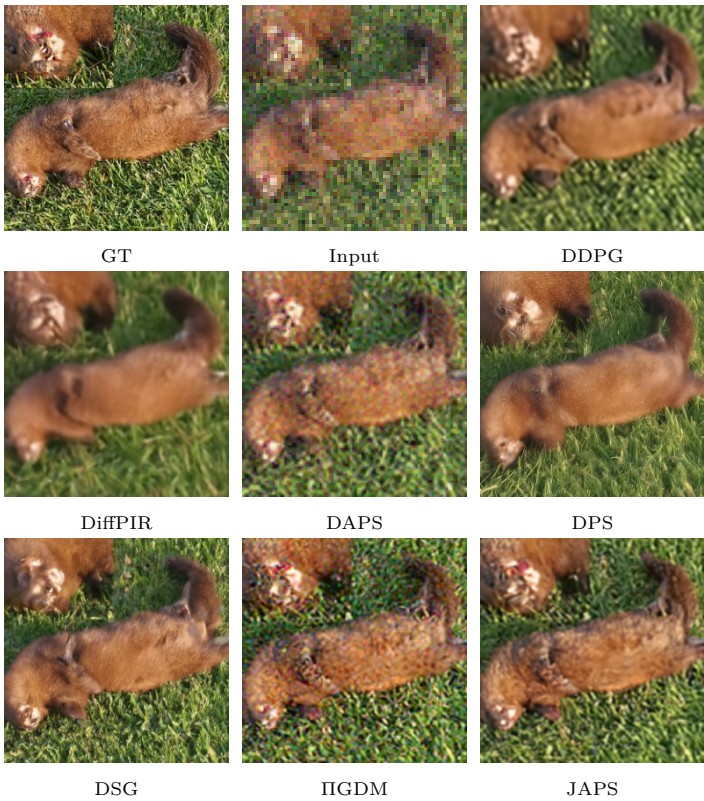

Figure 14: Super-Resolution results on ImageNet

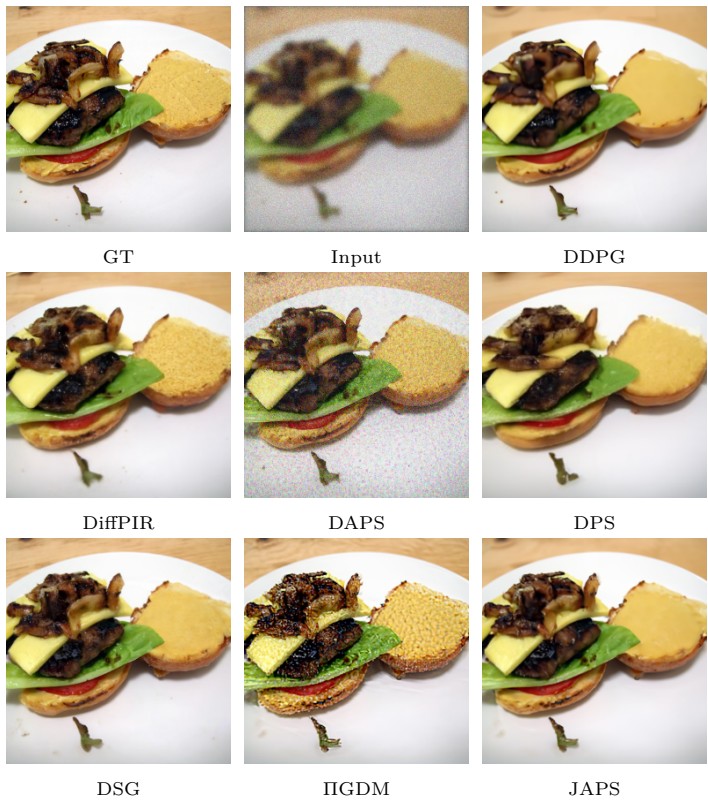

Figure 15: Gaussian Deblurring results on ImageNet

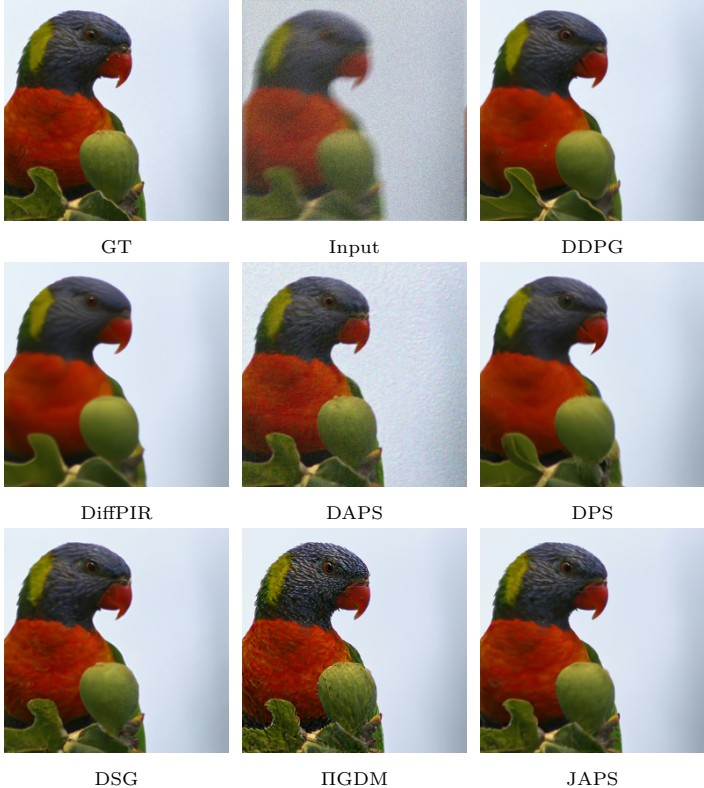

Figure 16: Motion Deblurring results on ImageNet

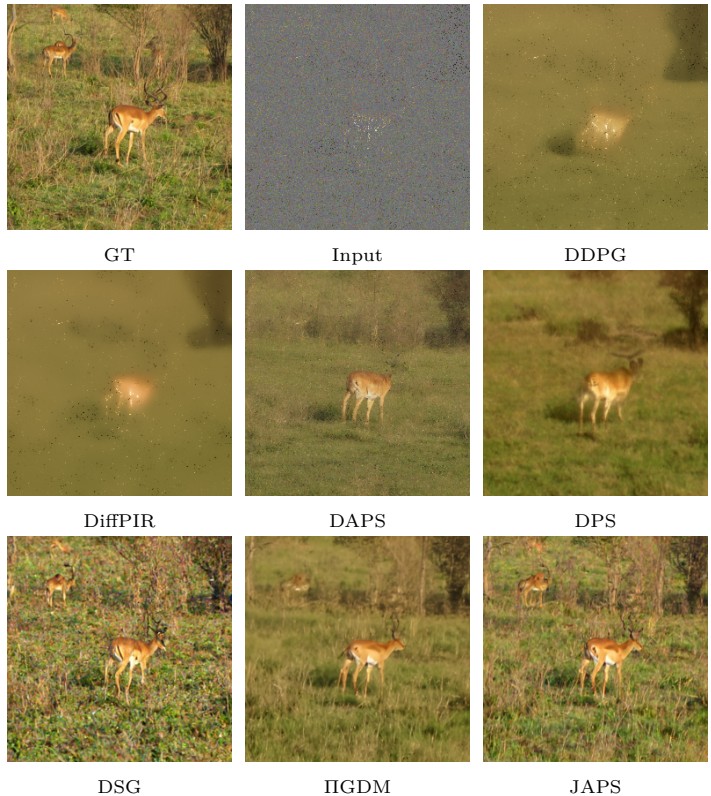

Figure 17: Random Inpainting results on ImageNet

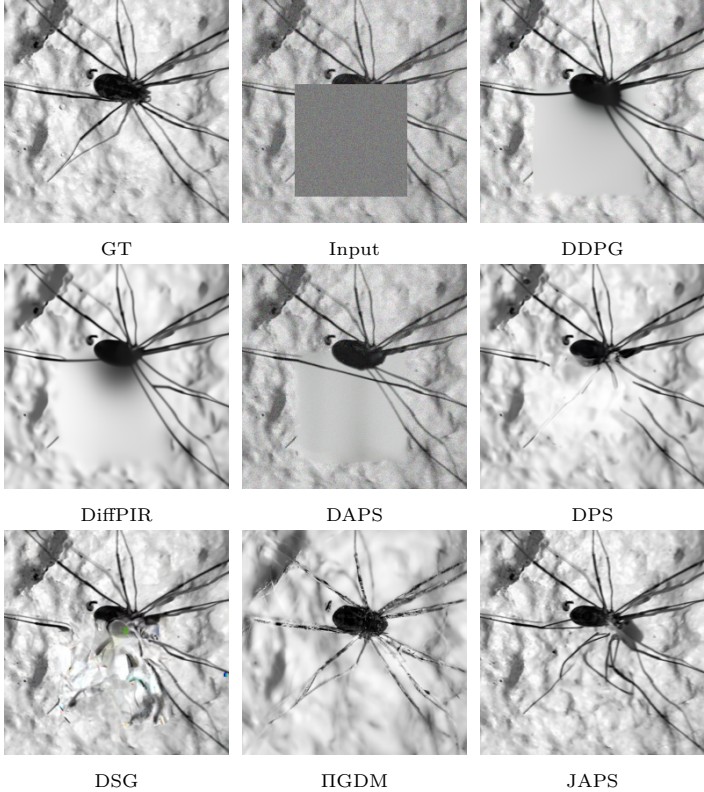

Figure 18: Box Inpainting results on ImageNet

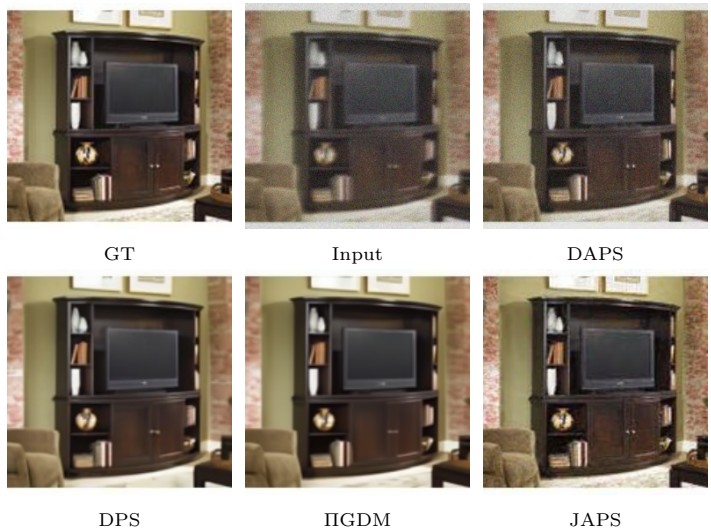

Figure 19: Nonlinear Blur results on ImageNet

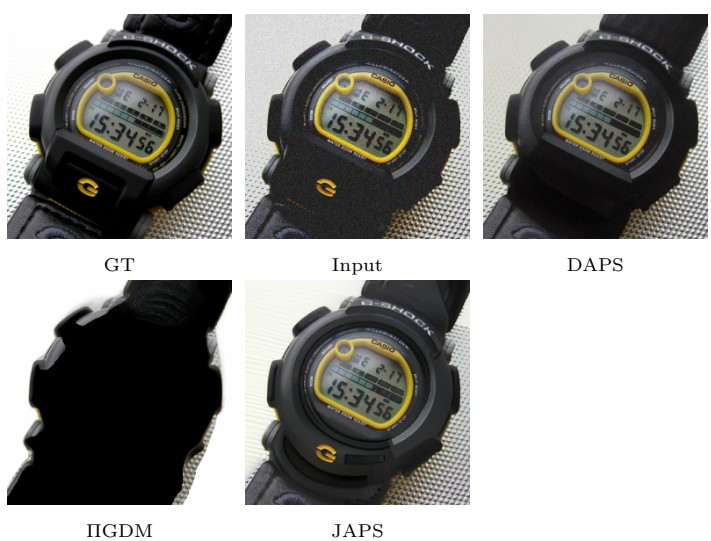

Figure 20: HDR results on ImageNet

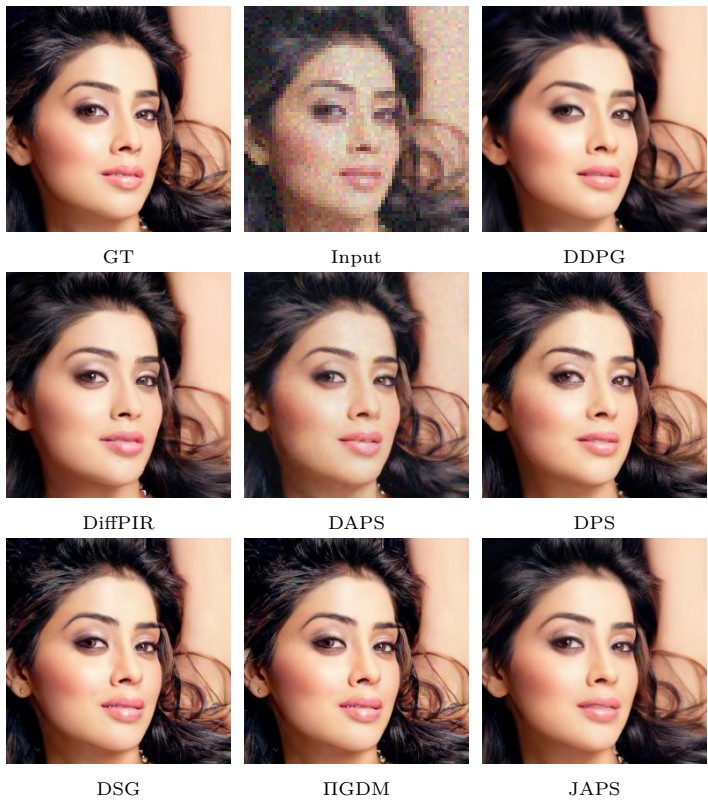

Figure 21: Super-Resolution results on CelebA-HQ

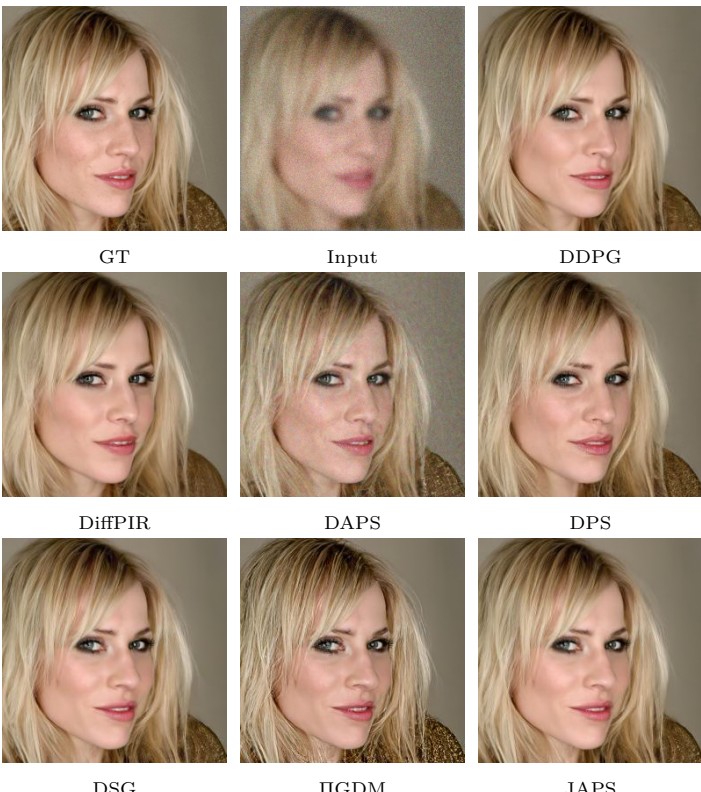

Figure 22: Gaussian Deblurring results on CelebA-HQ

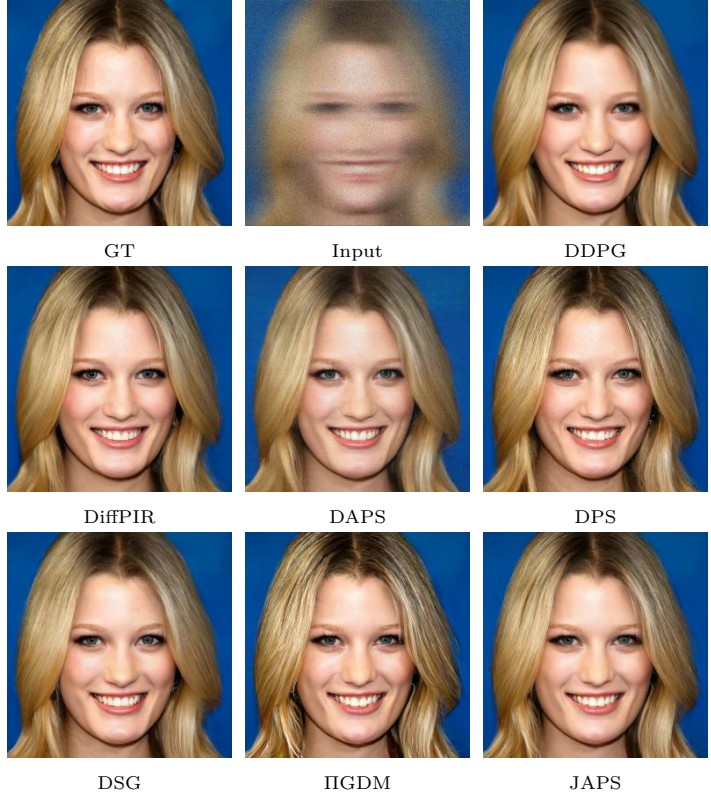

Figure 23: Motion Deblurring results on CelebA-HQ

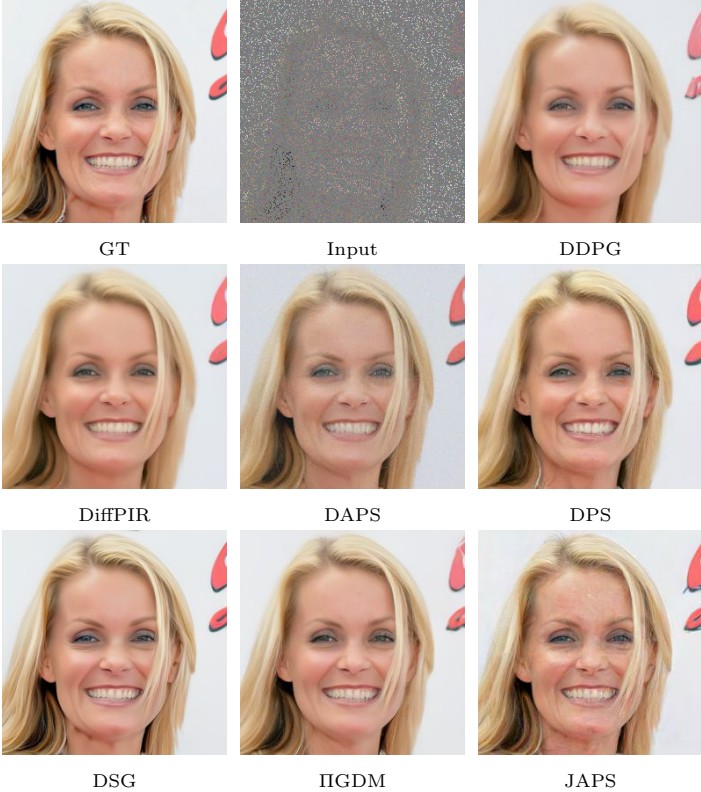

Figure 24: Random Inpainting results on CelebA-HQ

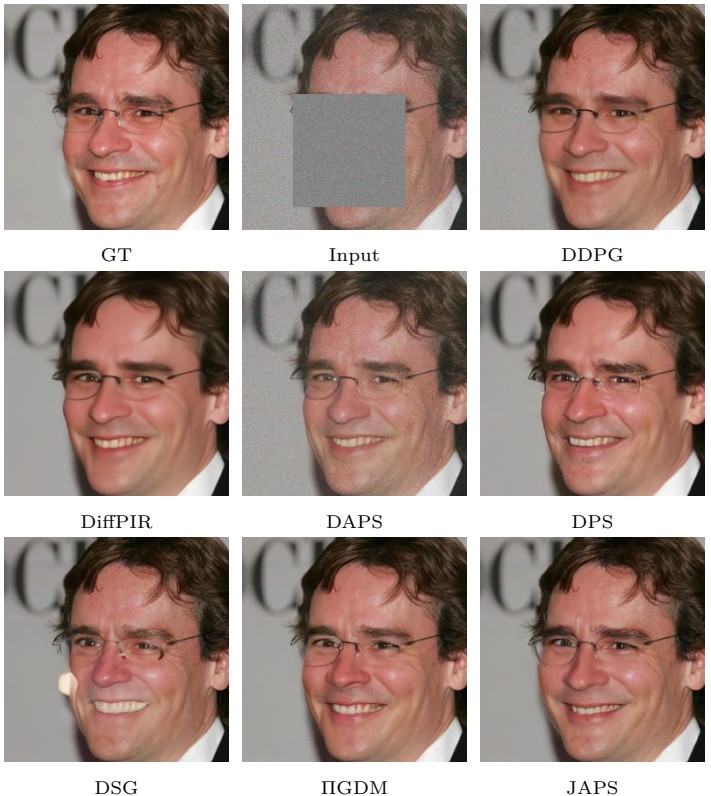

Figure 25: Box Inpainting results on CelebA-HQ

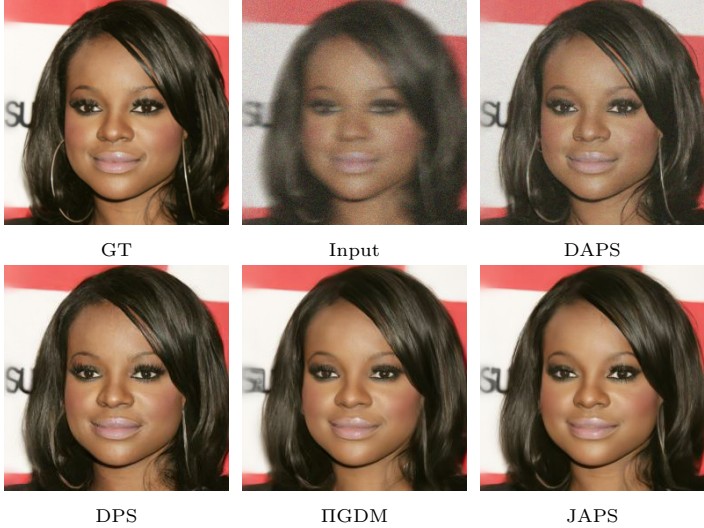

Figure 26: Nonlinear Blur results on CelebA-HQ

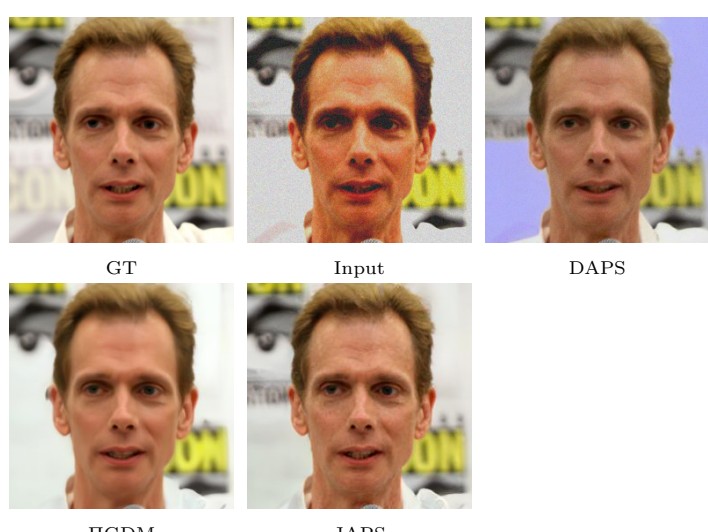

Figure 27: HDR results on CelebA-HQ

