# OpenReview forum: "Jacobian-Aware Posterior Sampling for Inverse Problems"
_TMLR — Accepted by TMLR_

### Review · Reviewer_6AUQ · 2026-04-04

**Summary Of Contributions:**

The paper observes that, in a Gaussian setting, direct and proximal diffusion posterior samplers depend on likelihood surrogates that differ only by the denoiser Jacobian. It then argues on a synthetic GMM that learned Jacobians can violate structural properties one would expect from the ideal Jacobian, namely symmetry and positive semidefiniteness. Based on this, the authors propose JAPS, which projects the proximal surrogate onto the span of the direct/Jacobian-aware surrogate, with the stated goal of retaining useful Jacobian information while being more robust to Jacobian error. The paper also gives a conditional DDIM reformulation with the missing time-dependent factor made explicit, extends the construction to non-linear operators via local linearization, and reports strong LPIPS/PSNR trade-offs on CelebA-HQ and ImageNet across several linear and non-linear tasks.

**Audience:**

Yes

**Audience Explanation:**

Yes. The paper is well-written, the taxonomy of methods is helpful, and the idea that the denoiser Jacobian is the key separator between direct and proximal formulations is useful. Even if one is not fully sold on JAPS itself, the paper provides a nice way to think about a currently fragmented literature.

**Broader Impact Concerns:**

None.

**Claims And Evidence:**

Yes

**Claims Explanation:**

The experimental evidence in the paper is broad, as it ranges over many linear and non-linear tasks over multiple datasets, together with ablation studies. The proposed method appears to be empirically useful.

The evidence that the Jacobian is not learned well in practice is weak: it is demonstrated only in a toy GMM, not on the actual image denoisers used in the main experiments.

**Requested Changes:**

Please provide a stronger justification for the projection rule in Eq. 10, which seems rather heuristic. In particular, can you verify experimentally that JAPS specifically mitigates learned Jacobian error?
- On the actual denoiser used in the image experiments, can you show that the Jacobian error is large?
- In the GMM setting where one can measure score error or sampling quality against ground truth, please directly compare direct, proximal, and JAPS guidance and check that the main reason for the JAPS improvement is due to Jacobian errors.

Can you compare JAPS against a simple linear interpolation between direct and proximal guidance?

Please separate the effects due to the JAPS projection, the corrected conditional DDIM factor, and the revised schedule.

The claim that JAPS comes at no additional computational cost seems strong; is this true even when compared against the cheaper proximal samplers?

Page 7: in the paragraph on baselines, “referred as” should be “referred to as”.

Appendix A.1, page 12: the sentence involving Tweedie’s formula is missing a word.

Appendix A.2, page 12: after Eq. 19, the minimizer is written as the expectation conditioned only on $\mathbf x_t$; it should be conditioned on both $\mathbf x_t$ and $\mathbf y$.

---

> ### Author Response · Authors · 2026-05-04
> **Official Comment to Reviewer 6AUQ**
>
> We thank the reviewer for the constructive feedback and for recognizing the value of our work. We have updated the manuscript to include real-world Jacobian measurements, synthetic ground-truth comparisons, and additional ablations to justify the JAPS projection rule.
>
> We reply by addressing the raised concerns point to point.
>
> ### **Stronger justification for the projection rule**
> We wish to emphasize that we do not claim theoretical optimality for our proposed solution. JAPS rather provides a practical solution that prioritizes simplicity and applicability by bridging the direct and proximal paradigms via projection.
> This choice is justified empirically in the ablation study in Section 4.3, as well as in new ablation studies presented in the subsequent points. Furthermore, we have added Section B.5 to the Appendix to specifically address the motivation for this projection rule. In brief, we find that limiting the search space of the optimal likelihood score surrogate $\mathbf{g}_t^*$ to the 1D span of $\mathbf{u}_t$ provides a principled regularization that preserves the essential prior information in $\mathbf{u}_t$ while yielding a computationally efficient and robust solution.
>
>
> ### **Verification of Jacobian Error in Real Denoisers**
> We have added measurements of the symmetry errors and the minimal eigenvalues of an off-the-shelf ImageNet denoiser to Section 3.2. The demonstrated properties of this real images’ Jacobian are similar to those of the GMM settings, suggesting the real-applied denoiser is also subject to the training-induced non-idealities that we bring up in this paper.
>
> ### **GMM comparison with Ground Truth**
> We added Section C.3 to the appendix. In this section, we show that while for an analytical denoiser $\mathbf{u}_t$ performs best, it diverges when using a trained denoiser, where JAPS performs best. Comparing JAPS to $\mathbf{u}_t$ and $\mathbf{v}_t$ directly isolates the impact of the Jacobian on the posterior sampling procedure, effectively demonstrating our main message: while the Jacobian exhibits useful information for posterior sampling, it also carries harmful effects when trained, which should be addressed and mitigated (e.g., via JAPS).. For full details, please refer to section C.3 in the revised paper.
>
>
> ### **Comparison to linear interpolation**
> We have added an experiment to the appendix (Section C.2) on SR$\times 4$. We construct a likelihood surrogate as a convex combination of the proximal and direct solutions and show that JAPS surpasses the best combination coefficient. This experiment showcases that a simple combination is not sufficient for extracting the most information out of the two solutions. Instead, our results highlight yet again that the Jacobian—and mainly its scaling properties—is harmful for constructing useful likelihood approximations, and the JAPS projection acts as a necessary regularizer.
>
>
> ### **Isolating the contribution of each component**
> Regarding the request to isolate the contribution of each component, we note that this is already presented in the manuscript. Our two main contributions are the elucidated treatment of the DDIM scheduling (the $\gamma_t$ factor) and the proposed guidance form.
>
> Scheduling: The experiments using $\mathbf{u}_t$ (Section 4.3) differ from $\Pi$GDM only by the scheduling time-dependent constants. Comparing $\mathbf{u}_t$ in Figure 3 and $\Pi$GDM on the same tasks, shows that our proposed schedule improves the posterior sampling, as the guidance vector between the two is nearly identical.
>
> Guidance form: The contribution of our specific guidance form is extensively studied in the ablation in Section 4.3.
>
>
> ### **Computational cost**
> We clarify that our method comes at no additional computational cost only compared to direct likelihood approximation methods that require backpropagating via the denoiser. This is now clarified this in the revised paper, Section 3.3.
> It is worth noting that while Jacobian-free proximal samplers are computationally cheaper, as they bypass the need for backpropagating via the denoiser, these methods rather require careful tuning to perform on par with direct methods (DDPG), may exhibit either degraded performance (DiffPIR) or require extensive sampling steps (DAPS).
>
> In addition to all, we are thankful for the minor fixes suggestions, and included them in the revised version of our work.
>
> We appreciate the constructive feedback used to strengthen our paper.

---

> > ### Comment · Reviewer_6AUQ · 2026-05-10
> >
> > Thank you for the response. I still believe that the lack of principled justification is a serious limitation, but the new empirical results address most of my main concerns.

---

### Review · Reviewer_2DjM · 2026-04-06

**Summary Of Contributions:**

This paper proposes Jacobian-aware posterior sampling (JAPS) for inverse problems. The authors identify that direct likelihood estimation and proximal solution can be written as closely related likelihood surrogate that differ by the denoiser Jacobian. Therefore, their proposed method incorporate both information to formulate a new likelihood estimate and plug into DDIM sampler for conditional posterior sampling. The paper also briefly discusses a intuitive extension of the method to nonlinear operators.

Strength:
1. The approach to unify direct and proximal methods by the denoiser Jacobian is insightful. The construction of paired surrogates make the comparison explicit.
2. The empirical experiments are fair and comprehensive. The method is compared to extensive recent posterior sampling methods and ablation studies are conducted to advocate the effectiveness of the proposed method.

Weakness:
1. The method is theoretically analyzed under strong isotropic Gaussian assumption on $p(x_0|x_t)$, which generally does not hold in real world given the complex data prior.
2. The theoretical result rely heavily on condition that MAP equals conditional mean, which again is a very strong condition. Outside this setting, the interpretation becomes heuristic.
3. The method that project likelihood estimation from proximal solution onto direct estimate is mainly heuristic without grounded mathematical evidence or some theoretical insight into this specific choice.
4. The extension to nonlinear measurement just plug in a linear approximation for the nonlinear operator and discarding higher order information. However, the derivation of proximal solution actually requires high-order information, so the extension to nonlinear case lack theoretical justification.
5. Empirical experiments lack notation of statistical significance, e.g. standard deviation.

**Audience:**

Yes

**Audience Explanation:**

Diffusion posterior sampling for inverse problems is a highly active topic, and the paper’s Jacobian-centric explanation of why direct and proximal guidance behave differently, and a practical method intended to mitigate Jacobian non-idealities without extra cost, should interest researchers in diffusion inference, inverse problems, and score-based sampling.

**Broader Impact Concerns:**

I do not identify significant broader impact or ethical concerns that require additional Broader Impact Statement.

**Claims And Evidence:**

Yes

**Claims Explanation:**

The paper’s central narrative that direct and proximal diffusion posterior samplers can be expressed as closely related likelihood surrogates differing mainly by the denoiser Jacobian, and combining these surrogates can improve conditional sampling are supported by explicit derivations and clear empirical evaluations across multiple inverse problems and baselines.

However, the claim that the method provide a mathematically grounded foundation for inverse problem solving is too strong. Several components that are central to the method rely on strong approximations or lack theoretical justification.
1. The isotropic Gaussian surrogate assumption for $p(x_0 \mid x_t)$.
2. Conditions under which the MAP solution equals the conditional mean (used to interpret a proximal/MAP solve as approximating $\mathbb{E}[x_0 \mid x_t,y]$).
3. The projection of $v_t$ onto $\mathrm{span}(u_t)$, which is motivated intuitively as mitigating Jacobian artifacts but is not accompanied by a deeper theoretical justification.
4. The nonlinear extension similarly uses a first-order linearization of the measurement operator, and its validity outside the locally linear regime is not theoretically characterized.

**Requested Changes:**

Critical:
1. Theoretically justify the method without strong isotropic Gaussian assumption or the condition that MAP equals conditional mean.
2. Provide theoretical justification of why the projection of $v_t$ onto $\mathrm{span}(u_t)$ will yield better result, especially the decrease in LPIPS.
3. Analyze extension to nonlinear measurement without discarding higher-order terms or analyze the approximation error when only consider linear approximation.
4. Conduct empirical experiemnts and report mean and std on the graph to check whether the method is statistically significant over other methods. The elimination of good performing methods with higher NFEs is unfair. Please conduct all experiements with 1000 NFEs and compare all methods.

Non-critical:
1. Notation error in A.2 after eq (19), $\mathbb{E}[x_0|x_t]$ should be instead $\mathbb{E}[x_0|x_t,y]$?
2. Proof in Proposition A.2, should show why you can interchange integral with gradient for eq (24)
3. Fix notation inconsistency (e.g. P16 stochasticity changes $\eta$ which does not appear in Section 2.2)
4. Structure theoretical results as theorems/lemmas/propositions and list them explicitly in the main paragraph and prove them in appendix instead of words like "For detailed derivations see ..." or "...(proof in ...)"

---

> ### Author Response · Authors · 2026-05-04
> **Official Comment to Reviewer 2DjM**
>
> We thank the reviewer for the examination of our theoretical framework. We acknowledge that our derivations rely on common approximations in the diffusion literature; however, we emphasize that our primary contribution lies in identifying and mitigating the structural non-idealities of the learned Jacobian. We have revised the paper to clarify these assumptions, tone down the claims where appropriate, and include the requested statistical analyses.
>
> We would like to address them point by point, starting from the critical points.
>
> ### **Isotropic Gaussian Assumption**
> This is a conventional and widely adopted approximation in the field to restore the tractability of the integral $p(\mathbf{y}|\mathbf{x}_t)$, as used in prominent methods like DPS, $\Pi$GDM, and TMPD. In the absence of a closed-form alternative for complex priors, this approximation serves as a necessary practical foundation for posterior sampling.
>
> ### **MAP-Mean coincidence**
> Under the isotropic Gaussian assumption and linear (or linearized) measurement operators, the equivalence of the MAP and the conditional mean is a direct consequence of the resulting Gaussian posterior, rather than an independent assumption.
>
> ### **Theoretical Justification of JAPS**
> We wish to clarify that while the search for a theoretically optimal likelihood surrogate for non-ideal backbones remains an open problem, our choice is a principled approach to structural regularization. As detailed in the new Section B.5, we use the 1D projection to preserve the Jacobian's directional guidance while using the measurement-consistent proximal solution $\mathbf{v}_t$ to regularize its magnitude.  This choice is justified empirically in Section 4.3 and the new study in Section C.2, which demonstrate that our projection outperforms alternative combinations and competing methods. We believe this framework provides a robust middle ground for leveraging trained Jacobians in posterior sampling, while we acknowledge that further theoretical exploration of this gap remains a promising direction for future research.
>
> ### **Higher-order Terms in Nonlinear Operators**
> We appreciate the opportunity to clarify our contribution regarding nonlinear operators. Many existing methods (e.g., DPS) do not differentiate between weakly and strongly nonlinear operators and often require a large number of diffusion steps. Our intention is to provide a "middle ground" that exploits local linearity for speed. In response to this concern, we have rephrased section 3.5 to focus the scope on weakly nonlinear operators, in which higher order terms are negligible rather than providing a general theoretical solution. Please also note that focusing the paper on linear operators is well justified by the diverse image processing tasks that they cover; accordingly, many important works have exclusively considered linearly modeled tasks.
>
> ### **Statistical Significance and NFE Counts**
>
> We have added section C.1 to our appendix, providing violin plots of our results. These plots provide not only a notion of mean performance with standard deviation, but also include a density estimation that informs other quantities such as long tailness and skewness. These plots demonstrate that JAPS not only improves average performance but also maintains a stable distribution without introducing high variance or long-tail artifacts compared to baselines.
>
> Regarding NFEs, We align our experiments with the current trend of reducing NFE requirements for generative and restoration tasks. Comparing at 1000 NFEs would move away from the practical regime our method targets. Moreover, most of the compared methods were specifically calibrated for 100 NFEs, so that reproducing them at 1000 NFEs would require careful calibration or fail (e.g. $\Pi$GDM).  In section C.4, however, we demonstrate a representative case in which increasing the NFEs improves the perceptual quality with a cost on distortion, both insignificantly, suggesting that our choice of 100 NFEs is satisfactory for our purposes of demonstrating our main message, which is the problematic usage of the Jacobian for posterior samplers, and our suggested mitigation.
>
>
> ### **[Non-criticals] Proofs and Formatting**
> We have addressed the requested technical corrections:
>
> Eq. 19: Fixed the conditioning notation to reflect $(\mathbf{x}_t, \mathbf{y})$.
>
> Proposition A.2: Added the derivation showing the validity of interchanging the integral and gradient.
>
> Structure: We have restructured the theoretical result regarding the relation between the posterior covariance and the Jacobian of the denoiser into an explicit proposition in the main text, due to its central role in our perspective of the importance of the Jacobian. However, we prefer keeping the proposition on the mean-mode coincidence in the Appendix as supplementary to the main message of this paper.
>
> We appreciate the constructive feedback used to strengthen our paper.

---

> > ### Comment · Reviewer_2DjM · 2026-05-11
> > **Further questions and comments**
> >
> > Thank you so much for the work and effort in addressing my concerns. I still have some questions and comments.
> >
> > 1. The motivation of your method comes from non-ideality of the Jacobian matrix. Why non-ideality of Jacobian harm posterior sampling?
> > 2. Your proposed likelihood surrogate still follows the direction of $u_t$, which contains the Jacobian. You have addressed in section B.5 that when symmetric part of $J_t$ is not p.s.d, it may reverse the sign, but after the reversion, the new direction may also not guarantee a p.s.d Jacobian and may even possibly make previously positive directions become negative (consider J_t contain both positive and negative eigenvalues). Why does the proposed likelihood surrogate help mitigate the issue of non-ideality of Jacobian (both p.s.d issue and non symmetric issue)?
> > 3. For statistical significance, my major concern is whether the performance of the new method is statistically significantly better compared with other method, so simply adding $mean \pm std$ in the tables is enough. The new violin plot does not provide clear visualization of this information.
> > 4. Concerns about NFE. If practical concern is the main target of your method, running all methods with 100 NFEs and compare your method with all other methods may provide stronger evidence of your method’s better performance rather than running some methods at 100 NFEs and others at 1000 NFEs while excluding those with more NFEs for comparison.
> > 5. In section 3.1, I don't see the formal mathematical definition of likelihood surrogate. It may be clearer to state in a proposition to first state the definition and then list the two computations for $v_t$ and $u_t$ with proofs in the appendix.
> >
> > Thank you so much for addressing my questions.

---

> > > ### Author Response · Authors · 2026-05-16
> > > **Response to Reviewer's 2DjM Further questions**
> > >
> > > Thank you for your reply. Addressing the questions point to point:
> > >
> > >
> > > 1. In Section 3.1 we show that the Jacobian is the key differentiator between direct and proximal likelihood score estimation, and in Subsection 3.1.1 we interpret it as a probabilistic precondioner of the likelihood ascent direction, but only for an ideal denoiser. By proving that the empirical Jacobians of trained denoisers systematically violate both symmetry and positive semi-definiteness (PSD) constraints (Section 3.2), we rule out the interpretation of the Jabobian as a probabilistic preconditioner. That is the reason that Jacobians of non-ideal denoisers harm posterior sampling.
> > > This analysis is validated empirically in Appendix Section C.3 (introduced during this revision cycle) within a controlled GMM environment where the ground-truth distribution and ideal denoiser are tractable. Our results demonstrate that while the direct likelihood approximation ($\mathbf{u}_t$) performs optimally under an ideal denoiser, its performance degrades significantly when utilizing a trained model - a failure mode successfully mitigated by JAPS.
> > >
> > >
> > > 2. Since symmetry and positive semi-definiteness (PSD) are necessary but not sufficient conditions for an ideal Jacobian, evaluating the mitigation of these specific geometric properties alone does not inherently guarantee better posterior sampling. We instead treat these structural violations as diagnostic evidence of non-ideality and focus our assessment on direct end-to-end performance improvements within Section C.3. Our experiments demonstrate that JAPS achieves the lowest $L_2$ distance to the ground-truth likelihood score when using a trained denoiser, implying an implicit mitigation of these underlying non-idealities. In the theoretical scenario where $c_t^* < 0$ (sign reversal), JAPS aligns the surrogate's direction to the half-space of the proximal estimate $\mathbf{v}_t$, which is mathematically guaranteed to provide a valid measurement-consistency ascent direction. In cases of poor directional alignment within the half space, the absolute magnitude of $c_t^*$ naturally minimizes to protect the update path, and such degenerate states are exceptionally rare during empirical sampling, as illustrated in Figure 5.
> > >
> > >
> > > 3. The violin plots provided in Figure 6 explicitly illustrate the statistical spread of our reconstruction results, demonstrating a distribution variance and stability comparable to the other baseline methods. For quantitative completeness, we provide the explicit mean and standard deviation specifications for these reported benchmarks in the following comment.
> > >
> > >
> > > 4. Regarding NFEs, all baselines were evaluated using their official code bases and the authors' optimized parameter configurations. Evaluating these methods outside their intended NFE regimes introduces severe performance mismatches: running 100-NFE methods over higher step counts ignores the original tuning and causes severe degradation (e.g., the performance drop of $\Pi$GDM shown in Section C.4) , while compressing 1000-NFE methods down to fewer steps triggers a similar decline (as documented in Section C.5 of [1], and I.1 of [2]). In contrast, JAPS explicitly incorporates step spacing to scale gracefully across varying step counts, making it natively robust and valid for comparison across both the 100 and 1000 NFE regimes. Furthermore, we do not omit 1000-NFE methods from our comprehensive evaluation; we exclude them strictly from the quantitative ranking to preserve a standardized computational baseline. As detailed in our wall-clock benchmarks in Section D.2, these 1000-NFE methods require massive computational overhead, demanding a $5.5\times$ to $7.5\times$ longer inference duration than JAPS on identical hardware.
> > >
> > >
> > > 5. The likelihood score is defined in Section 2.3 as $\nabla_{\mathbf{x}_t} \log p_t (\mathbf{y} \mid \mathbf{x}_t)$. A likelihood score surrogate is any function that aims to approximate the likelihood score, e.g. $\mathbf{u}_t$ from Eq. (9).
> > >
> > >
> > > Citations:
> > >
> > > [1]: *Chung, Hyungjin, et al. "Diffusion posterior sampling for general noisy inverse problems." arXiv preprint arXiv:2209.14687 (2022).*
> > >
> > > [2]: *Zhang, Bingliang, et al. "Improving diffusion inverse problem solving with decoupled noise annealing." Proceedings of the Computer Vision and Pattern Recognition Conference. 2025.*

---

> > > > ### Author Response · Authors · 2026-05-16
> > > > **Mean and standard deviations for ImageNet results**
> > > >
> > > > We share the results on ImageNet as in Table 1, together with the standard deviations over 1000 samples. The metrics are presented in the following convention, as in the main manuscript: PSNR / SSIM / LPIPS (excluding FID as it is not a sample-wise measure).
> > > >
> > > > | Method | Super Resolution | Gaussian Blur | Motion Blur | Random Inpainting | Box Inpainting |
> > > > | :--- | :--- | :--- | :--- | :--- | :--- |
> > > > | DDPG | 25.55 ± 3.08 / 0.722 ± 0.128 / 0.302 ± 0.117 | 27.74 ± 3.32 / 0.793 ± 0.108 / 0.170 ± 0.094 | 25.90 ± 3.23 / 0.735 ± 0.125 / 0.210 ± 0.113 | 21.85 ± 2.64 / 0.602 ± 0.154 / 0.395 ± 0.124 | 20.98 ± 3.29 / 0.749 ± 0.041 / 0.206 ± 0.046 |
> > > > | DiffPIR | 23.53 ± 2.73 / 0.643 ± 0.148 / 0.331 ± 0.121 | 24.56 ± 3.02 / 0.645 ± 0.158 / 0.192 ± 0.092 | 24.13 ± 3.24 / 0.665 ± 0.153 / 0.334 ± 0.139 | 22.27 ± 2.74 / 0.617 ± 0.157 / 0.376 ± 0.122 | 21.07 ± 3.56 / 0.761 ± 0.084 / 0.244 ± 0.076 |
> > > > | _DAPS_ | _23.60 ± 2.07 / 0.626 ± 0.090 / 0.422 ± 0.112_ | _21.74 ± 1.27 / 0.504 ± 0.085 / 0.331 ± 0.104_ | _25.14 ± 2.90 / 0.696 ± 0.119 / 0.240 ± 0.113_ | _22.80 ± 2.76 / 0.595 ± 0.125 / 0.309 ± 0.111_ | _20.26 ± 2.88 / 0.675 ± 0.047 / 0.248 ± 0.060_ |
> > > > | _DPS_ | _24.10 ± 3.09 / 0.655 ± 0.149 / 0.242 ± 0.114_ | _23.62 ± 3.11 / 0.629 ± 0.157 / 0.262 ± 0.115_ | _20.09 ± 4.58 / 0.503 ± 0.212 / 0.365 ± 0.193_ | _23.05 ± 2.93 / 0.646 ± 0.150 / 0.263 ± 0.108_ | _18.97 ± 3.24 / 0.750 ± 0.080 / 0.207 ± 0.069_ |
> > > > | _DSG_ | _24.33 ± 2.65 / 0.669 ± 0.124 / 0.203 ± 0.100_ | _26.69 ± 3.39 / 0.744 ± 0.126 / 0.153 ± 0.096_ | _24.33 ± 3.47 / 0.667 ± 0.143 / 0.197 ± 0.098_ | _21.91 ± 2.75 / 0.576 ± 0.123 / 0.236 ± 0.077_ | _19.18 ± 2.98 / 0.769 ± 0.055 / 0.151 ± 0.042_ |
> > > > | ΠGDM | 22.85 ± 2.65 / 0.595 ± 0.124 / 0.268 ± 0.105 | 22.83 ± 3.24 / 0.607 ± 0.144 / 0.227 ± 0.082 | 20.97 ± 3.21 / 0.528 ± 0.156 / 0.292 ± 0.101 | 23.74 ± 3.18 / 0.690 ± 0.136 / 0.231 ± 0.089 | 19.60 ± 3.18 / 0.788 ± 0.053 / 0.152 ± 0.040 |
> > > > | JAPS | 24.55 ± 3.06 / 0.687 ± 0.133 / 0.195 ± 0.093 | 27.16 ± 3.27 / 0.770 ± 0.113 / 0.149 ± 0.083 | 25.31 ± 3.38 / 0.713 ± 0.134 / 0.184 ± 0.096 | 23.71 ± 3.23 / 0.694 ± 0.126 / 0.168 ± 0.060 | 19.44 ± 3.02 / 0.785 ± 0.052 / 0.143 ± 0.037 |

---

> > > > > ### Author Response · Authors · 2026-05-16
> > > > > **Mean and standard deviations for CelebA-HQ results**
> > > > >
> > > > > We share the results on CelebA-HQ as in Table 1, together with the standard deviations over 1000 samples. The metrics are presented in the following convention, as in the main manuscript: PSNR / SSIM / LPIPS (excluding FID as it is not a sample-wise measure).
> > > > >
> > > > > | Method | Super Resolution | Gaussian Blur | Motion Blur | Random Inpainting | Box Inpainting |
> > > > > | :--- | :--- | :--- | :--- | :--- | :--- |
> > > > > | DDPG | 29.40 ± 1.46 / 0.838 ± 0.044 / 0.086 ± 0.025 | 30.42 ± 1.58 / 0.853 ± 0.042 / 0.054 ± 0.015 | 29.01 ± 1.78 / 0.827 ± 0.051 / 0.066 ± 0.021 | 25.63 ± 1.66 / 0.780 ± 0.058 / 0.146 ± 0.044 | 25.98 ± 3.17 / 0.852 ± 0.030 / 0.073 ± 0.028 |
> > > > > | DiffPIR | 26.58 ± 1.47 / 0.763 ± 0.064 / 0.081 ± 0.021 | 28.92 ± 1.61 / 0.813 ± 0.054 / 0.064 ± 0.017 | 27.34 ± 1.97 / 0.783 ± 0.066 / 0.078 ± 0.024 | 26.74 ± 1.62 / 0.808 ± 0.052 / 0.138 ± 0.039 | 25.88 ± 3.16 / 0.859 ± 0.035 / 0.096 ± 0.033 |
> > > > > | _DAPS_ | _28.12 ± 1.29 / 0.781 ± 0.045 / 0.075 ± 0.018_ | _28.34 ± 1.03 / 0.756 ± 0.034 / 0.091 ± 0.021_ | _28.42 ± 1.63 / 0.794 ± 0.050 / 0.072 ± 0.020_ | _26.65 ± 1.33 / 0.753 ± 0.041 / 0.120 ± 0.031_ | _24.14 ± 2.31 / 0.723 ± 0.035 / 0.165 ± 0.046_ |
> > > > > | _DPS_ | _27.56 ± 1.63 / 0.777 ± 0.062 / 0.072 ± 0.019_ | _28.28 ± 1.75 / 0.790 ± 0.061 / 0.065 ± 0.017_ | _26.28 ± 2.29 / 0.749 ± 0.077 / 0.083 ± 0.026_ | _26.77 ± 1.51 / 0.778 ± 0.058 / 0.085 ± 0.025_ | _26.79 ± 2.89 / 0.857 ± 0.034 / 0.064 ± 0.024_ |
> > > > > | DSG | 27.57 ± 1.46 / 0.780 ± 0.053 / 0.072 ± 0.016 | 30.29 ± 1.65 / 0.845 ± 0.045 / 0.051 ± 0.014 | 27.57 ± 3.22 / 0.790 ± 0.091 / 0.079 ± 0.056 | 28.39 ± 1.83 / 0.828 ± 0.056 / 0.079 ± 0.037 | 25.47 ± 3.53 / 0.855 ± 0.038 / 0.068 ± 0.039 |
> > > > > | ΠGDM | 27.23 ± 1.60 / 0.767 ± 0.061 / 0.078 ± 0.019 | 27.67 ± 1.99 / 0.778 ± 0.065 / 0.087 ± 0.025 | 26.15 ± 2.11 / 0.741 ± 0.076 / 0.104 ± 0.030 | 28.16 ± 1.73 / 0.827 ± 0.046 / 0.072 ± 0.020 | 27.11 ± 3.06 / 0.875 ± 0.030 / 0.050 ± 0.019 |
> > > > > | JAPS | 28.39 ± 1.63 / 0.806 ± 0.055 / 0.070 ± 0.019 | 30.25 ± 1.63 / 0.845 ± 0.045 / 0.050 ± 0.013 | 28.70 ± 1.98 / 0.814 ± 0.059 / 0.061 ± 0.019 | 27.90 ± 1.62 / 0.805 ± 0.043 / 0.069 ± 0.017 | 27.58 ± 2.77 / 0.874 ± 0.029 / 0.046 ± 0.017 |

---

### Review · Reviewer_TAgb · 2026-04-26

**Summary Of Contributions:**

# Summary
This paper proposes Jacobian-aware Posterior Sampling (JAPS), a diffusion-based method for zero-shot inverse problems. The main idea is that existing diffusion posterior samplers can be viewed as belonging to two broad families: direct likelihood-gradient methods, and proximal methods. The paper argues that the denoiser Jacobian contains useful local prior information, but that trained denoisers have non-ideal Jacobians that can be asymmetric or non-PSD. The paper introduces JAPS, which constructs a paired proximal surrogate and direct Jacobian-preconditioned surrogate, then projects the proximal direction onto the direct direction to keep useful Jacobian information while reducing harmful artifacts. The paper presents empirical validation on CelebA-HQ and ImageNet-256 against DAPS, DiffPIR, DDPG, DPS, DSG, and ΠGDM, with JAPS showing strong perceptual quality and competitive distortion metrics across several linear and nonlinear tasks.

# Theoretical framing.
The main strength of the paper is that it gives a clear conceptual explanation for the difference between direct and proximal diffusion inverse-problem solvers. The observation that these methods differ largely by whether the denoiser Jacobian appears in the likelihood surrogate is interesting and useful. I also found Proposition 3.1 helpful, since it expresses the likelihood score as a scaled difference between the conditional and unconditional posterior means. This gives a clean bridge between proximal estimation and likelihood-score guidance. The interpretation of the denoiser Jacobian as a posterior covariance-like preconditioner is also appealing and helps explain why the Jacobian might be useful rather than merely an implementation detail.

# Synthetic Jacobian Experiment.
The GMM experiment (section 3.2) analyzing trained Jacobian non-idealities shows that a trained denoiser can violate symmetry and PSD properties. This is interesting as it gives a concrete reason to be cautious about directly using the Jacobian in posterior sampling.

**Additional Comments:**

Overall, I think this is a strong and interesting paper. The connection between direct and proximal posterior sampling is interesting, the projection-based sampler is simple, and the experiments are good. My main concerns are that the key Jacobian non-ideality evidence is mostly synthetic, the projection rule needs more justification and edge-case analysis. Additionally, I think that the empirical claims would benefit from stronger uncertainty reporting, and clearer compute-normalized comparisons.

**Audience:**

Yes

**Audience Explanation:**

I think the paper is on topic and relevant.

**Broader Impact Concerns:**

None that I can think of.

**Claims And Evidence:**

Yes

**Claims Explanation:**

I think the paper has good empirical and theoretical validation.

**Requested Changes:**

# Proposed guidance rule.
The projection rule is simple and easy to implement. Projecting the proximal direction onto the Jacobian-preconditioned direction is an elegant way to combine the two sources of information, and the ablation study (Figure 3) is useful. I believe it would be useful if the authors explain better why projection onto the one-dimensional span of $u_t​$ is the best way to mitigate non-ideal Jacobians? I would also like to see more discussion of edge cases: what happens when the norm of $u_t$ is very small, or when
$\langle v_t,u_t\rangle$ is either negative, small or even (nearly) orthogonal?

# DDIM reformulation.

The conditional DDIM reformulation is a nice and practical contribution. The paper argues that many direct methods add guidance without correctly accounting for the DDIM time-step factor, and the ablation against ΠGDM across different numbers of sampling steps is convincing. This is a useful point independent of the projection idea. I would argue that this part of the paper could be improved from the writing perspective, there are several equations in the text and without proper names. This makes the reading more complicated than it should. I would suggest polishing the presentation of results.

# Experimental evaluation.

The experimental suite is reasonably broad: two datasets, several linear tasks, two nonlinear tasks, and multiple diffusion posterior-sampling baselines. The results are generally strong, especially for LPIPS, and the nonlinear inverse-problem results are particularly encouraging. However, some reported gains are small, and the paper does not report standard errors, confidence intervals, or paired significance tests. Could the authors present some information on the variance of the results?

# Perception-distortion tradeoff.

The paper frames JAPS as achieving a good balance between perceptual quality and distortion. However, some baselines have higher PSNR or SSIM, while JAPS tends to be strongest on LPIPS. This is not a weakness by itself, but the paper should be more careful with “state-of-the-art” phrasing. A Pareto-front presentation would be clearer than ranking each method by individual metrics.

# Fairness of baseline comparisons.

The baseline comparison is strong in coverage, but the NFE issue is somewhat confusing. Some methods are run with 1000 NFEs and excluded from ranking, while JAPS is run with 100 NFEs. This may be reasonable if those baselines require more steps, and it is a strength if JAPS is both better and cheaper. However, the comparison would be clearer with a more fair comparison say: explicit wall-clock time, memory usage, or NFE-normalized performance. If a method uses 10× more NFEs but has competitive quality, that is relevant; if JAPS is better at 100 NFEs and faster, that should be documented quantitatively. The claim of “no additional computational cost” should also be clarified: it appears to mean no additional cost beyond computing the direct Jacobian-preconditioned direction, not no additional cost relative to purely proximal methods that avoid backpropagation through the denoiser.

---

> ### Author Response · Authors · 2026-05-04
> **Official Comment to Reviewer TAgb**
>
> We thank the reviewer for the assessment of our framing and the GMM experiments. We have updated the manuscript to include a detailed edge-case analysis of the projection rule, polished the DDIM reformulation for clarity, and added comprehensive statistical and timing results.
>
> We reply by addressing the raised concerns point to point.
>
> ### **Proposed guidance rule**
> We would first like to clarify that we do not argue for the theoretical optimality of our proposed guidance rule. Instead, we use it to highlight the nature of the Jacobian—necessary yet potentially harmful—and offer a solution centered on simplicity and applicability. We empirically ground this choice through our ablation studies (Sections 4.3, C.2, and C.3), which show that our combined direct and proximal approach yields the best results among all tested methods. We believe the question of an "optimal" implementation for trained Jacobians remains an open gap for future investigation, motivated by our work.
> However, we agree that a discussion addressing edge cases is useful, and we have added such a discussion in Appendix B.5. There, we explain that edge cases where $c_t^* < 0$ or $c_t^* \approx 0$ reflect invalid states arising from non-ideal backbones that JAPS is designed to mitigate or suppress. As shown in our analysis, these states are atypical and do not occur in practice during standard sampling.
>
> ### **DDIM Reformulation**
> We appreciate the recognition of the conditional formulation of DDIM as an isolated contribution of our work. We have followed the reviewer’s advice and polished the corresponding section in our paper (3.4) by naming key equations in the text and improving overall flow.
>
> ### **Experimental Evaluation**
> We have addressed the request for uncertainty reporting by adding Section C.1 to the Appendix.We provide violin plots for all tasks that visualize the distribution of our results (PSNR and LPIPS) across the test sets. These plots demonstrate that JAPS not only improves average performance but also maintains a stable distribution without introducing high variance or long-tail artifacts compared to baselines.
>
> ### **Perception-distortion tradeoff**
> We agree that clear positioning is essential. We have clarified in the text that JAPS excels specifically in achieving state-of-the-art perceptual quality while maintaining competitive distortion metrics. We kindly refer the reviewer to Figure 1b, which presents a Pareto-style summary of the average PSNR/LPIPS trade-off across tasks, positioning JAPS relative to all other baselines.
>
> ### **Fairness of baseline comparisons**
> We have clarified the NFE and cost comparisons to provide a fairer picture:
>
> *Timing and NFEs:*  We have added Section D.2 and Table 4, which provide wall-clock time and NFE recordings for all methods. We note that while some baselines (like DSG or DAPS) can achieve high metrics at 1000 NFEs, they require significantly more compute time (e.g., 5.5x to 7.5x more than JAPS).
>
> *"No Additional Cost" Claim:* In response to this concern, we have clarified in Section 3.3 that JAPS comes at no additional cost relative to the direct method $\mathbf{u}_t$, as $\mathbf{v}_t$ is extracted from the same backpropagation graph for free. We do not claim that JAPS has less computational cost than existing proximal methods alternatives.
>
>
> We appreciate the constructive feedback used to strengthen our paper.

---

> > ### Comment · Reviewer_TAgb · 2026-05-18
> > **Reply**
> >
> > I thank the authors for taking care of my review with careful consideration to the points I made.
> > All of my comments have been taken care of, and I believe this revised version of the paper has increased the quality of the presentation significantly.

---

### Author Response · Authors · 2026-05-04
**General Response to All Reviewers**

We sincerely thank all reviewers for their careful reading of our manuscript and for the thoughtful, constructive feedback they provided. The comments have helped us substantially improve both the clarity and the technical contributions of the paper.


We have uploaded a revised version of the manuscript that addresses the points raised across the reviews. Modifications in the manuscript are teal-colored for ease of review.

In addition to this general response, we have posted detailed, point-by-point replies to each reviewer's comments separately.

Best regards,

The Authors

---

### Author Response · Authors · 2026-06-16
**Camera-Ready Submission**

We thank the AE for the constructive feedback and the decision to accept our manuscript.
We have updated the camera-ready version to address all of the requested clarifications. Specifically, we have:
- Clarified Assumptions: Explicitly stated our reliance on the linear-Gaussian surrogate framework in the Abstract, Introduction, and Conclusion.
- Refined Terminology: Corrected the phrasing regarding the Jacobian chain-rule application and properly qualified the MAP = conditional mean equivalence.
- Improved Precision: Clarified the wording in the nonlinear extension and renamed the statistical appendix to "Empirical distribution visualization."

Best regards,

The Authors.

---

### Decision · Action_Editor_ZJis · 2026-06-06

**Recommendation:** Accept with minor revision

**Audience:**

Yes

**Audience Explanation:**

A large portion of TMLR is interested in generative and conditional generative models. This paper will thus be of interest to a large portion of TMLR.

**Claims And Evidence:**

Yes

**Claims Explanation:**

This paper proposes a new diffusion-based posterior sampler for inverse problems by relating two families of existing methods, direct likelihood-gradient methods and proximal methods, through the denoiser Jacobian. The authors argue that the Jacobian contains useful local prior information, but that learned, practical denoisers result in imperfect Jacobians, so using this information directly can be harmful. Their method, JAPS, tries to keep some of the useful Jacobian-aware direction while regularizing it using the corresponding proximal update.

The reviewers generally agreed that the paper is relevant and that the Jacobian-based perspective is interesting. The main concerns were about the strength of the assumptions behind the derivation, the lack of a fully principled justification for the projection rule, the evidence that learned Jacobian errors are actually responsible for the observed behavior, the statistical significance of the empirical gains, and the fairness of the NFE comparisons.

The authors responded constructively. They added real-denoiser Jacobian diagnostics, additional synthetic GMM experiments where ground truth quantities are available, ablations against direct/proximal guidance and convex combinations, NFE comparisons, and additional statistical summaries. Two reviewers were satisfied or mostly satisfied after these changes. A third reviewer remained somewhat unconvinced, mainly pointing to the theoretical justification of the derived projection rule. I view this as a limitation in positioning, rather than a reason to reject the paper. I think this paper makes a neat observation that allows for practical gains and that can lead to future interesting directions. At the same time, I believe the concerns raised during the review process (as well as my own reading of the paper) suggest that some elements are overstressed, and that the paper has limitations in terms of positioning and phrasing. I therefore recommend acceptance, conditioned on a few clarifications.

The most important point is that the authors need to be much clearer about the Gaussian assumptions made in their derivation. The derivation relies on a per-time-step isotropic Gaussian surrogate for p(x_0| x_t), together with linear or locally linear measurements. I was surprised that this was not mentioned as a limitation, or as a direction for future work, in the conclusion. I agree with the authors that these assumptions are common in practical diffusion papers, but not all readers will be familiar with these. These assumptions should therefore be stated already in the abstract and made explicit in the introduction, so that the reader is not led to believe that the derivations and connections hold generally. Relatedly, the MAP = MMSE/conditional-mean statement should be qualified: this equivalence holds in the linear-Gaussian surrogate setting (which is a strong assumption) and not as a general property of posterior sampling with diffusion priors. Relatedly, the useful observation that direct and proximal methods differ by the presence of the denoiser Jacobian should be presented as being true only within the authors’ paired surrogate framework, rather than as a general equivalence across all direct and proximal methods.

Some wording and writing in the manuscript should be cleaned up for precision. In particular, (i) saying that the proximal solution “does not include the Jacobian” is confusing, since the solution is a point (so 'inclusion' here is imprecise). The distinction is really that direct likelihood surrogates acquire a denoiser-Jacobian factor through the chain rule, while proximal surrogates do not apply that factor. Likewise, (ii) the phrase “Jacobian attachment” should also be avoided or rephrased. 'Attachment' is not standard terminology and obscures the intended meaning. (iii) The nonlinear extension should be described more precisely. The authors write "For a nonlinear forward operator A(·), the linear-Gaussian assumption is violated, causing the MAP estimate to deviate from the true conditional mean". This could be read as conveying that in the linear case, the posteriors are Gaussian. This is only true if one assumes them to be so (and it is not due to the operator being linear). Lastly, (iv) I agree with Rev 2DjM that the added section on “Assessment of Statistical Significance” does not, as it stands, constitute any validation of statistical significance. It is a valuable experimental addition, but no hypothesis testing is performed to actually assess statistical significance. For simplicity, this section should just be renamed as "Empirical distribution visualization" or something to that effect (unless the authors want to actually include paired inferential analysis or hypothesis testing).

By and large, this is a nice paper. My recommendation is that the authors make these clarifications to let the true contributions of their work shine, and not be cluttered by a lack of clarity of some statements.